# DOUBLE DYNAMIC SPARSE TRAINING FOR GANS

## ABSTRACT

The past decade has witnessed a drastic increase in modern deep neural networks (DNNs) size, especially for generative adversarial networks (GANs). Since GANs usually suffer from high computational complexity, researchers have shown an increased interest in applying pruning methods to reduce the training and inference costs of GANs. Among different pruning methods invented for supervised learning, dynamic sparse training (DST) has gained increasing attention recently as it enjoys excellent training efficiency with comparable performance to post-hoc pruning. Hence, applying DST on GANs, where we train a sparse GAN with a fixed parameter count throughout training, seems to be a good candidate for reducing GAN training costs. However, a few challenges, including the degrading training instability, emerge due to the adversarial nature of GANs. Hence, we introduce a quantity called balance ratio (BR) to quantify the balance of the generator and the discriminator. We conduct a series of experiments to show the importance of BR in understanding sparse GAN training. Building upon single dynamic sparse training (SDST), where only the generator is adjusted during training, we propose double dynamic sparse training (DDST) to control the BR during GAN training. Empirically, DDST automatically determines the density of the discriminator and greatly boosts the performance of sparse GANs on multiple datasets.

## 1 INTRODUCTION

In the past decade, the training and inference costs of modern deep neural networks (DNNs) are gradually becoming prohibitive (He et al., 2016; Dosovitskiy et al., 2020; Liu et al., 2021d), especially for large language models (Brown et al., 2020). Among all these large models, generative adversarial networks (GANs) (Goodfellow et al., 2020) have been widely investigated for years and achieved remarkable results. However, similar to other giant models, GANs are notably computationally intensive. For example, BigGAN (Brock et al., 2018) trained on 8 NVIDIA V100 GPUs with full precision will take 15 days. Consequently, to train GANs in broader resource-constrained scenarios, this computational bottleneck of training needs to be resolved urgently.

Neural network pruning has recently emerged as a powerful tool to reduce training and inference costs of DNNs for supervised learning. There are mainly three genres of pruning methods, namely pruning-at-initialization, pruning-during-training, and post-hoc pruning methods. Post-hoc pruning (Janowsky, 1989; LeCun et al., 1989; Han et al., 2015) can date back to the 1980s, which was first introduced for reducing inference time and memory requirements; hence does not align with our purpose of efficient training. Later, pruning-at-initialization (Lee et al., 2018; Wang et al., 2020a; Tanaka et al., 2020) and pruning-during-training methods (Louizos et al., 2017; Wen et al., 2016) were introduced to prune the networks before training and throughout the training, respectively. Most early pruning-during-training algorithms (Savarese et al., 2020) gradually decrease the density of the neural networks and hence do not bring much training efficiency compared to post-hoc pruning. However, recent advances in dynamic sparse training (DST) (Evci et al., 2020; Liu et al., 2021a;b;c; Mocanu et al., 2018) for the first time show that pruning-during-training methods can have comparable training FLOPs as pruning-at-initialization methods while having competing performance with respect to post-hoc pruning. Therefore, applying DST on GANs seems to be a promising choice.

Although DST has attained remarkable achievements in supervised learning, the application of DST on GANs is less explored due to newly emerged challenges. The main difficulty stems from the fact that the training procedure of GANs is notoriously brittle. To ensure successful training, we

usually need carefully chosen architectures and finely-tuned hyper-parameters. One possible cause is the difficulty of balancing the generator and the discriminator throughout training (Bai et al., 2018; Arora et al., 2017). Specifically, an overly-strong discriminator will lead to overfitting, while a weak discriminator will result in mode collapse. As a consequence, the requirement of balanced training brings even more challenges to sparse GAN training. On the one hand, we find that performance degradation caused by the unbalance of GANs is even more severe when sparsity is introduced. On the other hand, for directly applying DST to the generator (or both) like the pioneering work STU-GAN (Liu et al., 2022), it is unclear how to determine a reasonable density of the discriminator. To this end, we propose a metric called balance ratio (BR), which measures the degree of balance of the two components, to study sparse GAN training.

We find that BR is useful in (1) understanding the interaction between the discriminator and the generator, (2) identifying the cause of training failure, and (3) helping stabilize sparse GAN training as an indicator. To our best knowledge, this is the first study to investigate the balance of sparse GANs and may even provide new insights into dense GAN training.

Using BR as an indicator, we further propose double dynamic sparse training (DDST) to adjust the density and the connections of the discriminator automatically during training.

Our contributions are summarized below:

- We introduce a quantity named balance ratio to quantify the degree of balance in GAN training, which also helps understand the cause of some training failure cases.

- We first consider single dynamic sparse training (`SDST`), which is a generalization of STU-GAN (Liu et al., 2022): applying DST to only the generator with varying discriminator density ratios. We show that `SDST` does not necessarily outperform the static sparse training baselines.

- We provide two strategies to determine the discriminator density for `SDST`, and we find that using a relatively larger density usually generates stable and better performance.

- Using the balance ratio as an indicator, we propose double dynamic sparse training (`DDST`), which makes the discriminator dynamic both in density level and parameter level. Empirically, `DDST` outperforms baselines with reasonable computational cost on several datasets.

## 2 RELATED WORKS

### 2.1 NEURAL NETWORK PRUNING

Based on the smallest granularity of pruned units, neural network pruning can be categorized into structured (Liu et al., 2017; 2018; Huang & Wang, 2018; Luo et al., 2017) and unstructured pruning (Frankle & Carbin, 2018; Han et al., 2015). In this work, we mainly focus on unstructured pruning where individual weight is the finest resolution.

**Post-hoc pruning.** Post-hoc pruning prunes weights of a fully-trained neural network, and they usually have high computation cost due to the multiple rounds of train-prune-retrain procedure (Han et al., 2015; Renda et al., 2020). Some use specific criteria (Han et al., 2015; LeCun et al., 1989; Hassibi et al., 1993; Molchanov et al., 2019; Dai et al., 2018; Guo et al., 2016; Dong et al., 2017; Yu et al., 2018) to remove weights, while others perform extra optimization iterations (Verma & Pesquet, 2021). Post-hoc pruning was initially proposed to reduce the inference time, while later work on lottery ticket works (Frankle & Carbin, 2018; Renda et al., 2020) aimed to mine trainable sub-networks.

**Pruning-at-initialization methods.** SNIP (Lee et al., 2018) is one of the pioneering works which aim to find trainable sub-networks without any training. Some following works (Wang et al., 2020a; Tanaka et al., 2020; de Jorge et al., 2020; Alizadeh et al., 2022) aim to propose different metrics to prune networks at initialization. Among them, Synflow (Tanaka et al., 2020), SPP (Lee et al., 2019), and FORCE (de Jorge et al., 2020) try to address the problem of layer collapse during pruning. Neural tangent transfer (Liu & Zenke, 2020) learns a sub-network by aligning the empirical neural tangent kernel and network output to the dense counterpart.

**Pruning-during-training methods.** Another genre of pruning algorithms gradually prunes dense DNNs throughout training. To mitigate performance drop after pruning, early works add explicit $\ell_0$ (Louizos et al., 2017) or $\ell_1$ (Wen et al., 2016) regularization terms to encourage sparse solution. Later works learn the subnetworks structures through projected gradient descent (Zhou et al., 2021) or trainable masks (Kang & Han, 2020; Kusupati et al., 2020; Liu et al., 2020; Savarese et al., 2020; Srinivas et al., 2017; Xiao et al., 2019). However, these pruning-during-training methods often do not enjoy memory sparsity during training. As a remedy, DST methods (Bellec et al., 2017; Dettmers & Zettlemoyer, 2019; Evci et al., 2020; Liu et al., 2021a;b;c; Mocanu et al., 2018; Mostafa & Wang, 2019; Graesser et al., 2022) were introduced to train the neural networks under a given parameter budget while mask change is allowed during training.

## 2.2 GENERATIVE ADVERSARIAL NETWORKS

**Generative adversarial networks (GANs).** GANs (Goodfellow et al., 2020) have drawn considerable attention and have been widely investigated for years. Various architectures have been proposed to enhance the capability of GANs. Deep convolutional GANs (Radford et al., 2015) replace fully-connected layers in the generator and the discriminator. After that, follow-up works (Brock et al., 2018; Gulrajani et al., 2017; Karras et al., 2017; Zhang et al., 2019) employed more advanced methods to improve the fidelity of generated samples. Due to the difficulty of finding Nash Equilibrium, training of GAN is highly unstable. Therefore, several novel loss functions (Mao et al., 2017; Arjovsky et al., 2017; Salimans et al., 2016; Gulrajani et al., 2017; Sun et al., 2020), normalization and regularization methods (Miyato et al., 2018; Wu et al., 2021; Terjék, 2019) were proposed to stabilize the adversarial training. Besides the efforts devoted to the training of GAN, image-to-image translation is also extensively explored. Specifically, this direction includes semantic image synthesis (Zhu et al., 2017b), style transfer (Karras et al., 2020b; Choi et al., 2018; Zhu et al., 2017a), super resolution (Ledig et al., 2017; Wang et al., 2018) etc.

**GAN compression and pruning.** Like other deep neural networks, the training and inference process of GANs requires massive resource consumption and memory. One of the promising ways is based on neural architecture search and distillation algorithm (Li et al., 2020; Fu et al., 2020; Hou et al., 2021). Another part of the work applied prune-based methods for GANs' generator compression (Shu et al., 2019; Jin et al., 2021; Yu & Pool, 2020). Yet, they only focus on the pruning of generators, thus potentially posing a negative influence on Nash Equilibrium between generators and discriminators. Later, works by (Wang et al., 2020b) presented a unified framework by combing the methods mentioned above. Follow-up work by Li et al. (2021) compresses both components of GANs by letting the student GANs also learn the losses. Another line of work (Kalibhat et al., 2021; Chen et al., 2021) tries to test the existence of lottery tickets in GAN. However, most mentioned methods are not prepared for training efficiency and require over-parameterized GAN models in advance. Directly training sparse GANs has been less explored so far. To the best of our knowledge, STU-GAN Liu et al. (2022) is the only work that tries to apply DST to GANs.

# 3 BALANCE RATIO: QUANTIFYING THE BALANCE OF SPARSE GANS

## 3.1 PRELIMINARY AND SETUPS

Generative adversarial networks (GANs) have two fundamental components, a generator $G(\cdot; \boldsymbol{\theta}_G)$ and a discriminator $D(\cdot; \boldsymbol{\theta}_D)$. Specifically, the generator maps a sampled noise $\boldsymbol{z}$ from a multivariate normal distribution $p(\boldsymbol{z})$ into a fake image to cheat the discriminator, whereas the discriminator distinguishes the generator's output and the real images $\boldsymbol{x}_r$ from the distribution $q(\boldsymbol{x})$. Formally, the optimization objective of the two-player game defined in JS-GAN (Goodfellow et al., 2020) is defined as follows:

$$\mathcal{L}(\boldsymbol{\theta}_D, \boldsymbol{\theta}_G) = \mathbb{E}_{\boldsymbol{x}_r \sim q(\boldsymbol{x})} \left[ \log(D(\boldsymbol{x}_r; \boldsymbol{\theta}_D)) \right] + \mathbb{E}_{\boldsymbol{z} \sim p(\boldsymbol{z})} \left[ \log(1 - D(G(\boldsymbol{z}; \boldsymbol{\theta}_G))) \right]. \tag{1}$$

To be more specific, different loss can be used, including Wasserstein loss (Gulrajani et al., 2017) and hinge loss (Miyato et al., 2018). In this work, we use hinge loss for all GANs.

**GAN sparse training.** In this work, we are interested in sparse training for GANs. Specifically, the objective of sparse GAN training can be formulated as:

$$\min_{\boldsymbol{\theta}_G} \max_{\boldsymbol{\theta}_D} \mathcal{L}(\boldsymbol{\theta}_D \odot \boldsymbol{m}_D, \boldsymbol{\theta}_G \odot \boldsymbol{m}_G) \tag{2}$$

$$\text{s.t.} \quad \boldsymbol{m}_D \in \{0,1\}^{p_D}, \quad \boldsymbol{m}_G \in \{0,1\}^{p_G}, \quad \|\boldsymbol{m}_D\|_0/p_D \le d_D, \quad \|\boldsymbol{m}_G\|_0/p_G \le d_G,$$

where $\odot$ is the Hadamard product; $\boldsymbol{\theta}_D$, $\boldsymbol{m}_D$, $p_D$, $d_D$ are the sparse solution, mask, number of parameters, and target density for the discriminator, respectively. The corresponding variables for the generator are denoted with subscript $G$. For pruning-at-initialization methods, masks $\boldsymbol{m}$ are determined before training whereas $\boldsymbol{m}$ are dynamically adjusted for dynamic sparse training (DST) methods.

## 3.2 Balance of GAN during training

As discussed in Section 1, it is essential to maintain the balance of generator and discriminator during GAN training. As pointed out by Bai et al. (2018) and Arora et al. (2017), discriminators that are too strong lead to over-fitting, whereas weak discriminators are unable to detect mode collapse. When it comes to sparse GAN training, the consequences caused by the unbalance can be further amplified. Specifically, different from dense GAN training, densities of generators and discriminators can be varied significantly, leading to a more unbalanced worst-case scenario.

To support our claim, we conduct experiments with SNGAN (Miyato et al., 2018) on the CIFAR-10 dataset. Following Liu et al. (2022), we start with **static sparse training** where densities of generators and discriminators are chosen from $\{10\%, 20\%, 30\%, 50\%, 100\%\}$. Layer-wise sparsity ratio and masks $\boldsymbol{m}_G, \boldsymbol{m}_D$ are determined using *Erdős-Rényi-Kernel* (ERK) graph topology (Evci et al., 2020) and are fixed throughout the training. More experiment details can be found at Appendix A.

**Experiment results.** Results are reported in Figure 2. First three plots in Figure 2 show the results when varying density of discriminator $d_D$ for weak generators ($d_G \in \{10\%, 20\%, 30\%\}$). We observe FID first decreases then increases. Specifically, neither overly-weak discriminators nor overly-strong discriminators can provide satisfactory performance. Similarly, for stronger generators ($d_G \in \{50\%, 100\%\}$), only the dense discriminator with $d_G = 100\%$ is not too weak to have satisfactory FID result. Hence, to ensure a balanced training of GAN, it is crucial to find the suitable sparsity ratio for the discriminator.

## 3.3 Balance ratio

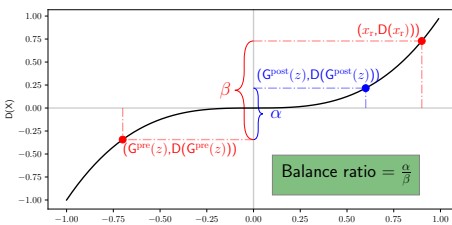

Figure 1: Illustration of balance ratio.

The observation in Section 3.2 raises a fundamental question: *is there a way to quantify the degree of balance between the generator and the discriminator?* To answer the question, we introduce balance ratio (BR), which is, to the best of our knowledge, the first quantity that measures the balance of sparse generators and discriminators.

At each training iteration, we draw random noise $\boldsymbol{z}$ from a multivariate normal distribution and real images $\boldsymbol{x}_r$ from the training set. We denote the discriminator after gradient descent update as $D(\cdot; \boldsymbol{\theta}_D)$. We denote generator before and after gradient descent training as $G^{\text{pre}}(\cdot; \boldsymbol{\theta}_G)$ and $G^{\text{post}}(\cdot; \boldsymbol{\theta}'_G)$, respectively. Then the balance ratio is defined as:

$$\text{BR} = \frac{D(G^{\text{post}}(\boldsymbol{z})) - D(G^{\text{pre}}(\boldsymbol{z}))}{D(\boldsymbol{x}_r) - D(G^{\text{pre}}(\boldsymbol{z}))} = \frac{\alpha}{\beta}. \tag{3}$$

Figure 1 also provides an illustration of BR. Specifically, BR measures how much improvement the generator can achieve in the scale measured by the discriminator for a specific random noise $\boldsymbol{z}$. When BR is small (e.g., BR$< 30\%$), it means that the updated generator is too weak to trick the discriminator as the generated images are still considered fake. Similarly, for the case where BR is large (e.g., BR$> 80\%$), the discriminator is considered too weak hence it will not provide useful information to the generator.

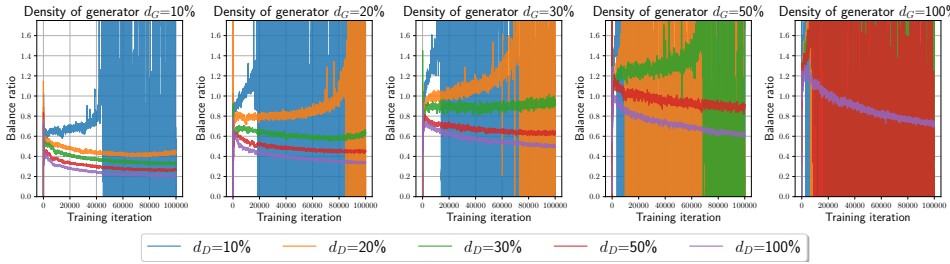

Figure 2: FID ($\downarrow$) of `static` sparsely trained SNGAN with and without `DA` on CIFAR-10 with different sparsity ratio combinations. The shaded areas denote the standard deviation.

Figure 3: Balance ratio of `static` sparsely trained SNGAN on CIFAR-10 with different sparsity ratio combinations.

We now visualize the BR evolution throughout the training for the experiments in Section 3.2. The results are shown in Figure 3.

**The effectiveness of BR.** We first check if BR can reflect the density increase (hence representation power increase) of the discriminator. In Figure 3 we can see that for larger discriminator density $d_D$, the BR is much lower throughout the training. Furthermore, the best density as indicated by Figure 2 has overall BR in the range $[0.3, 0.7]$.

**Overly weak discriminators lead to training failure.** For the cases where the discriminators are too weak compared to the generators, e.g., all cases where $d_D = 10\%$, we are able to observe the strongly oscillatory behavior of BR. More precisely, BR starts to oscillate after it reaches a value that is higher than 1.0. During the experiments, we also empirically observe that the FID gradually increases after such a turning point. As also shown in Figure 2, FID of overly-strong discriminators (e.g., $d_D = 100\%$) are lower than overly-weak discriminators (e.g., $d_D = 10\%$). The such phenomenon seems to imply that performance degradation caused by overly-strong discriminators is better than failure caused by overly-weak discriminators.

### 3.4 DYNAMIC DENSITY ADJUSTMENT OF THE DISCRIMINATORS

We have shown in Section 3.3 that BR is able to capture the degree of balance of the generators and discriminators. Hence, it is natural to leverage BR to dynamically adjust the density of discriminators during sparse GAN training. Specifically, we initialize the initial density of the discriminator $d_D^{init} = d_G$. After a specific training iteration interval $\Delta T_D$, we will adjust the density of the discriminator based on the time-averaged BR over last a few iterations with a pre-defined density increment $\Delta d$. With a pre-defined BR bounds $[B_-, B_+]$, we decrease $d_D$ by $\Delta d$ when BR is smaller than $B_-$, and vise versa. Notice that the `DA` algorithm is in spirit very similar to StyleGAN2-ADA (Karras et al., 2020a) which adjusts augmentation probability with ADA. Out of simplicity, we increase the density by growing the connections with the largest gradient magnitude (Evci et al., 2020). Global magnitude pruning is used to drop connections so as to decrease the density. The algorithm is shown in Appendix C Algorithm 1.

We test our proposed methods **dynamic density adjust (DA)** with two target BR intervals, namely `DA-strong` ($[B_-, B_+] = [0.3, 0.4]$), `DA-mild` ($[B_-, B_+] = [0.45, 0.55]$). `DA-strong` tends to find a relatively stronger discriminator, which results in a lower BR throughout the training, whereas `DA-mild` tends to make the discriminator and the generator relatively balanced, i.e., BR $\approx 0.5$.

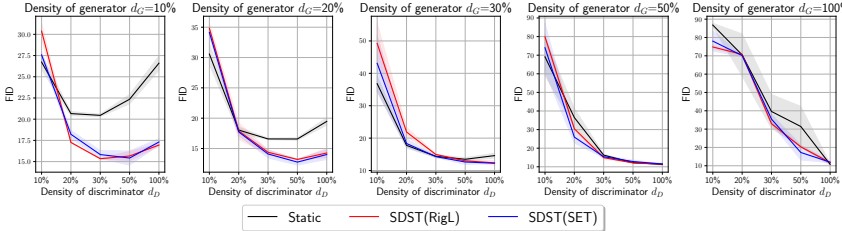

Figure 4: FID ($\downarrow$) comparison of `SDST` against `static` sparse training for SNGAN on CIFAR-10 with different sparsity ratio combinations. The shaded areas denote the standard deviation.

**Experiment results.** Results are shown in Figure 2 with dashed lines. For stronger generators ($d_G \in \{30\%, 50\%, 100\%\}$), both `DA-strong` and `DA-mild` are able to identify reasonable discriminator densities. While for weak generators ($d_G \in \{10\%, 20\%\}$), `DA-mild` shows a more stable performance. The experiments show the significant benefits brought by BR. Furthermore, they again support our claims that neither overly-strong nor weak discriminators can lead to balanced and successful GAN training.

## 4    IS ONLY ADJUSTING THE GENERATOR ENOUGH FOR SPARSE GANS?

In this section, we are going to test DST on GANs. We first test `SDST`, a direct application of DST method on GAN where only the generator dynamically adjusts masks during the training. We do not consider naively applying DST on both generators and discriminators, as in STU-GAN (Liu et al., 2022), it is empirically shown that adjusting both components simultaneously generates worse performance with more severe training instability. We name such method **single dynamic sparse training (SDST)** as only one component of the GAN, i.e., the generator, is dynamic. Hence, STU-GAN is a special case for `SDST`, which grows connections based on gradients.[1]

We follow the same setting considered in Section 3.2 where the densities of the generators $d_G$ and discriminators $d_D$ are chosen from $\{10\%, 20\%, 30\%, 50\%, 100\%\}$. Detailed DST procedure and corresponding hyper-parameters can be found in Appendix B.

**Experiment results.** We show the experiment results in Figure 4. The first observation is that the performance of ● `RigL` and ◆ `SET` does not vary much in general. The second observation is that `SDST` is better than static sparse training when the discriminator is strong enough. More specifically, for $d_G \in \{10\%, 20\%\}$, `SDST` method is worse than static sparse training when the density of the discriminator is weak ($d_D = 10\%$). On the contrary, when the discriminator is strong enough, $d_D \in \{20\%, 30\%, 50\%, 100\%\}$, we see a great performance boost brought by `SDST`. The reason is that the in-time over-parameterization induced by DST increases the representation power of the generator. Such a boost is beneficial only when the discriminator has matching or better representation power.

Despite the superior performance of STU-GAN (or `SDST` in general) at higher density ratios, there exist some limitations for `SDST`, which are summarized as follows:

❶ When using `SDST`, $d_D$ is manually chosen before training. However, it is unclear what is a good choice. In real-world scenarios, it is not practical to search for the optimal $d_D$ for each $d_G$.

❷ The issue of GAN unbalance is unresolved during training since the density of the discriminator is fixed. As shown in Figure 4, the best performance is not always obtained with the maximum $d_D = 100\%$. If we are using an overly-strong discriminator, we are wasting extra computational cost for a worse performance.

Hence, STU-GAN (or `SDST` in general), which directly applies DST to the generator, may only be useful when the corresponding discriminator is strong enough. In this sense, to deal with more complicated scenarios, obtaining balanced training in an automatic way is essential in GAN dynamic sparse training.

---

[1]Notice that STU-GAN is almost identical to ● `SDST` (`RigL`) with EMA tailored for DST.

Table 1: FID ($\downarrow$) of different sparse training methods on CIFAR-10 and STL-10 datasets with no constraint on the density of the discriminator. Best results are in **bold**; second-best results are underlined.

| | CIFAR-10 | | | | STL-10 | | | |
|---|---|---|---|---|---|---|---|---|
| Generator density | 10% | 20 % | 30 % | 50 % | 10% | 20 % | 30 % | 50 % |
| (Dense Baseline) | 10.74 | | | | 29.71 | | | |
| Static-Balance | 26.73 | 18.04 | 14.38 | 12.22 | 50.08 | 44.19 | 43.96 | 37.21 |
| Static-Strong | 26.60 | 19.47 | 14.60 | 11.28 | 52.03 | 44.04 | 42.53 | 38.33 |
| ◆ DST-balance (SET) | 32.02 | 18.54 | 14.74 | 13.23 | 49.91 | 33.71 | 32.92 | 31.75 |
| ● DST-balance (RigL) | 24.56 | 15.53 | 13.62 | 12.51 | 66.90 | 50.34 | 44.57 | 32.63 |
| ◆ SDST-Balance (SET) | 27.80 | 18.13 | 14.15 | 12.32 | 63.57 | 49.05 | 43.74 | 31.29 |
| ◆ SDST-Strong (SET) | 17.04 | 14.58 | 12.29 | 11.47 | 78.34 | 54.31 | 41.77 | 32.32 |
| ● SDST-Balance (RigL) | 30.38 | 17.89 | 14.95 | 12.09 | 46.17 | 38.12 | **31.88** | 31.30 |
| ● SDST-Strong (RigL) | 16.95 | 14.26 | 12.36 | 11.47 | 48.04 | 34.24 | 32.67 | **30.40** |
| ◆ R-DDST (SET) | **13.58** | 12.54 | 11.71 | **10.97** | 63.59 | 56.15 | 45.48 | 31.71 |
| ● R-DDST (RigL) | 13.77 | **12.33** | **11.46** | 11.18 | **42.72** | **33.12** | 32.44 | 30.88 |

Table 2: FID ($\downarrow$) and normalized training FLOPs of different sparse training methods with BigGAN on CIFAR-10 dataset. Best results are in **bold**; second-best results are underlined.

| | FID ($\downarrow$) | | | | Normalized training FLOPs | | | |
|---|---|---|---|---|---|---|---|---|
| Generator density | 10% | 20 % | 30 % | 50 % | 10% | 20 % | 30 % | 50 % |
| (Dense Baseline) | 8.43 | | | | $6.80 \times 10^{17}$ (100%) | | | |
| Static-Balance | 17.46 | 13.13 | 10.90 | 8.54 | 9.78% | 19.04% | 28.68% | 49.12% |
| Static-Strong | 22.96 | 13.54 | 11.54 | 9.02 | 83.90% | 84.93% | 86.32% | 90.15% |
| ● SDST-Balance (RigL) | 11.98 | 9.58 | 8.96 | **7.92** | 9.91% | 19.41% | 28.90% | 48.38% |
| ● SDST-Strong (RigL) | 10.79 | 9.30 | 8.82 | 8.30 | 84.04% | 85.22% | 86.54% | 89.56% |
| ● R-DDST (RigL) | **9.58** | **8.77** | **8.11** | 8.17 | 9.77% | 24.85% | 40.00% | 77.13% |

# 5 DOUBLE DYNAMIC SPARSE TRAINING FOR GANs

STU-GAN (or `SDST` in general) considered in the last section cannot generate stable and satisfying performance. This implies that we should utilize the discriminator in a better way rather than just directly applying DST to the discriminator. Consequently, `DA` (Section 3.4), which adjusts the discriminator density to stabilize GAN training, is a favorable candidate to address the issue. We name the proposed method **double dynamic sparse training (DDST)**, which adjusts the density of the discriminator during training with BR as the indicator while the generator performs DST. We propose two `DDST` methods, namely `R-DDST` and `S-DDST` based on whether we give constraints on the maximum density of the discriminator. We present them in Section 5.1 and Section 5.2. We use the word **double** for the following two reasons: ❶ both the generators and the discriminators are dynamic (both `R-DDST` and `S-DDST`); ❷ the discriminator enjoys two levels of dynamic flexibility, namely density level and parameter level (`S-DDST`). Such a method has much more flexibility and generates more stable performance compared to `SDST`.

## 5.1 RELAXED DOUBLE DYNAMIC SPARSE TRAINING

We first investigate the direct combination of `SDST` with `DA`. Specifically, the generator is adjusted using `SDST` as mentioned in Section 4 while the density of the generator is dynamically adjusted with `DA` as mentioned in Section 3.4. We call such a combination **relaxed double dynamic sparse training (R-DDST)** as it does not necessarily introduce sparsity to the discriminator, and the density of the discriminator can be as high as 100% (hence dense discriminator). For a fair comparison, baseline methods can use the discriminator with arbitrary sparsity ratio, i.e., $d_D \in [d_{\min}, d_{\max}] = [0\%, 100\%]$.

**Comparison to STU-GAN (SDST)**. Compared to STU-GAN (or `SDST` in general) which predefines the discriminator density before training, the difference is that for `R-DDST`, the density of the discriminator is adjusted during the training process automatically through `DA`. Given the initial discriminator density $d_D^{\text{int}} = d_G$, `R-DDST` automatically increases the discriminator density if a stronger discriminator is needed, and vice versa.

**Datasets, architectures, and target sparsity ratios.** We first conduct experiments on SNGAN with ResNet architectures on CIFAR-10 (Krizhevsky et al., 2009) and STL-10 (Coates et al., 2011) datasets. Target density ratios of the generators $d_G$ are chosen from $\{10\%, 20\%, 30\%, 50\%\}$. Please see Appendix A for more experiment details.

**Baseline methods and R-DDST.** We use `static` (Section 3.2) and SDST (Section 4) as our baselines. Since these baselines use pre-defined discriminator density ratios, we propose two strategies to define the discriminator density ratios based on the results from Section 3.2 and Section 4: ❶ balance strategy, where we set the density of the discriminator $d_D$ the same as the density of the generator $d_G$; ❷ strong strategy, where we set the density of the discriminator as large as possible, i.e., $d_D = d_{\max}$. In this section, we use $d_{\max} = 100\%$ for the strong strategy. For SDST methods, we test both grow methods, i.e., ◆ SDST(SET) (Mocanu et al., 2018) which grows connections randomly and ● SDST(RigL) (Evci et al., 2020) which grows connections via gradient.

Similar to SDST, we again consider ◆ R-DDST(SET) and ● R-DDST(RigL) which differ based on how R-DDST grows connections. One thing to notice is that we use the same growth criterion for the generator and the discriminator out of simplicity. More training details can be found in Appendix B. FID results on the training set are shown in Table 1. More results of SNGAN on CIFAR-10 test set can be found in Appendix E. Training costs comparison can be found in Appendix G.

**The strong strategy and the balance strategy.** For almost all density ratios of SNGAN (CIFAR-10) experiments, using the strong strategy is always comparable to or better than the balance strategy. The difference between the two is almost negligible when applied to `static` methods. However, for SDST methods, using stronger discriminators always leads to a large performance gain.

For SNGAN on the STL-10 dataset, the advantage of the strong strategy over the balance strategy is no longer obvious. Precisely, for 3 out of 8 cases, using the strong strategy is better than using the balance strategy. The explanation is that the size difference between generators and discriminators is larger for STL-10. Hence, the degree of unbalance is more severe and leads to more detrimental effects.

**R-DDST identifies reasonable discriminator density.** For the CIFAR-10 dataset, we find that R-DDST consistently performs better than the corresponding baselines with the same grow methods. This illustrates that R-DDST is flexible and able to find suitable discriminator density compared to the two baseline strategies, i.e., the strong and the balance strategy.

For the STL-10 dataset, ● R-DDST(RigL) performs consistently better than ◆ R-DDST(SET) and baselines, whereas ◆ R-DDST(SET) is not competitive any more. We postulate that under such a setting where the dataset scales up and the training is more difficult, gradient growth not only identifies important connections of the generator but also provides efficient representation power growth of the discriminator to balance the growth of the generator. Please also see Appendix D for the time evolution of BR and the discriminator density during training for R-DDST methods.

**Larger GAN model experiments.** We have also conducted experiments with BigGAN (Brock et al., 2018) on CIFAR-10 datasets. Based on the SNGAN results, we compare all ● RigL variants with `static` baselines. FID and normalized training FLOPs with respect to dense training are shown in Table 2. The results show that ● R-DDST shows stable performance and outperforms other baselines most of the time. Moreover, compared to the second best method ● SDST-Strong, ● R-DDST not only shows lower FID but also requires much less training cost.

**Main takeaway.** In this section, we compared R-DDST with sparse training baselines. We find that ● RigL and strong strategy are preferred compared to ◆ SET and balance strategy. ● SDST(RigL) with strong strategy generally generates better performance compared to other sparse training baselines. Finally, ● R-DDST(RigL) beats ● SDST(RigL) with much less computational cost and always ranked top two among all methods.

## 5.2 STRICT DOUBLE DYNAMIC SPARSE TRAINING

R-DDST introduced in the previous section does not necessarily introduce sparsity for the discriminator, which provides less memory/training resources saving for larger generator density ratios. Hence, we further present **strict double dynamic sparse training (S-DDST)** in this section which enforces the discriminator to be sparse, i.e., $d_D \leq d_{\max} < 100\%$. In this section, we assume that

Table 3: FID ($\downarrow$) of different sparse training methods on CIFAR-10 and STL-10 datasets. The density of the discriminator is constrained to be lower than 50%. Best results are in **bold**; second-best results are underlined.

| | CIFAR-10 | | | | STL-10 | | | |
|---|---|---|---|---|---|---|---|---|
| Generator density | 10% | 20 % | 30 % | 50 % | 10% | 20 % | 30 % | 50 % |
| (Dense Baseline) | 10.74 | | | | 29.71 | | | |
| Static-balance | 26.73 | 18.04 | 14.38 | 12.22 | 50.08 | 44.19 | 43.96 | 37.21 |
| Static-strong | 22.35 | 16.57 | 13.47 | 12.22 | 50.28 | 44.95 | 42.12 | 37.21 |
| ◆ DST-balance (SET) | 32.02 | 18.54 | 14.74 | 13.23 | 49.91 | 33.71 | 32.92 | 31.75 |
| ● DST-balance (RigL) | 24.56 | 15.53 | 13.62 | 12.51 | 66.90 | 50.34 | 44.57 | 32.63 |
| ◆ SDST-balance (SET) | 27.80 | 18.13 | 14.15 | 12.32 | 63.57 | 49.05 | 43.74 | 31.29 |
| ◆ SDST-strong (SET) | 16.00 | 13.31 | 13.17 | 12.32 | 48.40 | 33.56 | 32.19 | 31.29 |
| ● SDST-balance (RigL) | 30.38 | 17.89 | 14.95 | 12.09 | 46.17 | 38.12 | 32.48 | 31.30 |
| ● SDST-strong (RigL) | 15.66 | 13.20 | 12.99 | 12.09 | 63.65 | 33.45 | 32.09 | 31.30 |
| ◆ S-DDST (SET) | 14.22 | 13.30 | 12.39 | **11.97** | 51.72 | 35.74 | 42.36 | 31.68 |
| ● S-DDST (RigL) | **14.13** | **12.87** | **12.15** | 12.17 | **44.28** | **32.84** | **32.00** | **30.28** |

we can use the discriminator with sparsity ratio $d_D \in [d_{\min}, d_{\max}] = [0\%, 50\%]$. Compared with R-DDST, the learning process will be more challenging with the introduced constraints on the maximal discriminator density. S-DDST consists of two phases and works as follows:

❶ **Density exploration of the discriminator.** During the first phase, S-DDST performs just like R-DDST, with the exception that we apply the constraint $d_D \leq d_{\max} < 100\%$. Concretely, S-DDST aims to find a suitable discriminator density $d_D^*$ with DA algorithm in the first half of training.

❷ **Paramter exploration of the discriminator.** During the second phase, both the generator and discriminator are adjusted using DST with fixed discriminator density $d_D^*$ found in the first phase.

**Baseline methods and S-DDST.** We use the same baselines and adopt the same general setup in Section 5.1. We divide the training iterations evenly for two phases. For a comprehensive comparison, we continue to report FID results from two growth methods, i.e., ◆ S-DDST(SET) and ● S-DDST(RigL) in Table 3. IS and other results can be found in Appendix E.

**S-DDST shows stable and superior performance.** For the CIFAR-10 dataset, we notice that S-DDST stably surpasses its corresponding baselines regardless of grow methods and initial density of discriminators and generators. Even with a further constraint on the discriminator, DA is still able to improve GANs training and can explore more reasonable density than the strong and the balance strategy. For STL-10 dataset, ● S-DDST(RigL) again shows the most promising performance. Please also see Figure 7 in Appendix D for discriminator density and BR evolution during training.

**Main takeaway.** In this section, we report the results from ● S-DDST(RigL) with its baselines. Generally, ● RigL still demonstrates encouraging results compared with ◆ SET in most experiments when extra sparsity is introduced. While the strong strategy shows favorable performance in the CIFAR-10 dataset, the gain is not salient when the size of the backbone increase and the training dataset scales up to STL-10. Most importantly, ● S-DDST(RigL) is able to have comparable performance in some cases when compared to ● R-DDST(RigL) and outperforms ● SDST(RigL) after we restrict the maximal density of discriminators.

## 6 CONCLUSION

In this paper, we study DST for GANs. We find that simply applying DST methods to the generator is not sufficient to improve the performance of sparse GANs. Hence, we propose to use BR to measure the degree of unbalance between the generator and the discriminator. We find that the application of DST only on the generator is beneficial when the discriminator is relatively stronger. Furthermore, we propose two methods, namely R-DDST and S-DDST, to dynamically adjust the discriminator in both parameter and density levels. Both of these methods demonstrate encouraging results. Our study may help researchers have a better understanding of the balance of GAN training and encourage more researchers to investigate sparse training for generative models.

## 7 REPRODUCIBILITY STATEMENT

To ensure reproducibility, we will include a link to an anonymous repository after the discussion forums are open. All the experiment details can be found in Section 4, Section 5.1, Section 5.2, Appendix A and Appendix B.

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

## A  EXPERIMENTAL SETUP

Our code is mainly based on the original code of ITOP (Liu et al., 2021c) and GAN ticket (Chen et al., 2021).

### A.1  ARCHITECTURE DETAILS

We use ResNet-32 (He et al., 2016) for CIFAR-10 dataset and ResNet-48 for STL-10 dataset. See Table 4 and Table 5 for detailed architectures. Spectral normalization is applied for all fully-connected layers and convolutional layers of the discriminators.

For BigGAN architecture, we use the implementation used in Zhao et al. (2020). [2]

### A.2  DATASETS

We use the training set of CIFAR-10 and unlabeled partition of STL-10 for GAN training. Training images are resized to $32 \times 32$ and $48 \times 48$ for CIFAR-10 and STL-10 datasets, respectively. Augmentation methods for both datasets are random horizontal flip and per-channel normalization.

### A.3  TRAINING HYPERPARAMETERS

**SNGAN on the CIFAR-10 and STL-10 datasets.** We use a learning rate of $2 \times 10^{-4}$ for both generators and discriminators. The discriminator is updated five times for every generator update. We adopt Adam optimizer with $\beta_1 = 0$ and $\beta_2 = 0.9$. The batch size of the discriminator and the generator is set to be 64 and 128, respectively. Hinge loss is used following Brock et al. (2018); Chen et al. (2021). We use exponential exponential moving average (EMA) (Yaz et al., 2018) with $\beta = 0.999$. The generator is trained for a total of 100k iterations.

**BigGAN on the CIFAR-10 dataset.** We use a learning rate of $2 \times 10^{-4}$ for both generators and discriminators. The discriminator is updated four times for every generator update. We adopt Adam optimizer with $\beta_1 = 0$ and $\beta_2 = 0.999$. The batch size of both the discriminator and the generator is set to be 50. Hinge loss is used following Brock et al. (2018); Wu et al. (2021). We use EMA with $\beta = 0.9999$. The generator is trained for a total of 200k iterations.

### A.4  EVALUATION METRIC

**SNGAN on the CIFAR-10 and the STL-10 datasets.** We compute Fréchet inception distance (FID) and Inception score (IS) for 50k generated images every 2000 iterations. Best FID and IS are reported. For the CIFAR-10 dataset, we report both FID for the training set and test set, whereas, for the STL-10 dataset, we report the FID of the unlabeled partition.

**BigGAN on the CIFAR-10 dataset.** We compute Fréchet inception distance (FID) and Inception score (IS) for 10k generated images every 5000 iterations. Best FID and IS are reported.

## B  DYNAMIC SPARSE TRAINING DETAILS

### B.1  GENERAL DST HYPERPARAMETERS

Following Evci et al. (2020), we specify the hyper-parameters of DST through sparsity distribution, update schedule, drop criterion, and grow criterion. We explain the details of DST below.

**Sparsity distribution at initialization.** Following Evci et al. (2020); Liu et al. (2021c), only parameters of fully connected layers and convolutional layers will be pruned. At initialization, we use the commonly adopted *Erdős-Rényi-Kernel* (ERK) strategy (Evci et al., 2020; Dettmers & Zettlemoyer, 2019; Liu et al., 2021c) to allocates higher sparsity to larger layers. Specifically, the sparsity of convolutional layers $l$ is scaled with $1 - \frac{n^{l-1}+n^l+w^l+h^l}{n^{l-1}n^l w^l h^l}$, where $n^l$ denotes the number of channels

---

[2] `https://github.com/mit-han-lab/data-efficient-gans/tree/master/DiffAugment-biggan-cifar`.

of layer $l$ while $w^l$ and $h^l$ are the width and the height of the corresponding kernel in that layer. For fully connected layers, *Erdős-Rényi* (ER) strategy is used, where the sparsity is scaled with $1 - \frac{n^{l-1}+n^l}{n^{l-1}n^l}$.

**Drop and grow.** After $\Delta T$ training iterations, we update the mask $\boldsymbol{m}_G$ by dropping/pruning $f_{\text{decay}}(\gamma, T)p_G d_G$ number of connections with the lowest magnitude, where $p_G$, $d_G$ are the number of parameters and target density for the generator, $f_{\text{decay}}(\gamma, T)$ is the update schedule, which will be explained in the next part. Right after the connection drop, we regrow the same amount of connections.

For the growing criterion, we test both random growth ◆ SDST(SET) (Mocanu et al., 2018; Liu et al., 2021c) and gradient-based growth ● SDST(RigL) (Evci et al., 2020). Concretely, gradient-based methods find newly-activated connections $\theta$ with highest gradient magnitude $\left|\frac{\partial \mathcal{L}}{\partial \theta}\right|$, while random based methods explore connections in a random fashion. All the newly-activated connections are set to 0. One thing that should be noticed is that while previous works consider layer-wise connections drop and growth, we grow and drop connections globally as it grants more flexibility to the SDST method.

**Update schedule.** The update schedule can be specified by the number of training iterations between sparse connectivity updates $\Delta T$, the initial fraction of connections adjusted $\gamma$, and decaying schedule $f_{\text{decay}}(\gamma, T)$ for $\gamma$.

**EMA for sparse GAN.** EMA (Yaz et al., 2018) is well-known for its ability to alleviate the non-convergence of GAN. We also implement EMA for sparse GAN training. Specifically, we zero out the moving average of dropped weights whenever there is a mask change.

### B.2 DST HYPERPARAMETERS FOR SDST

**SNGAN on the CIFAR-10 and the STL-10 datasets.** The connection update frequency of the generator $\Delta T$ is set to 500 and 1000 for the CIFAR-10 dataset and STL-10 dataset, respectively. The initial $\gamma$ is set to 0.5 and we use a cosine annealing function $f_{\text{decay}}$ following RigL and ITOP.

**BigGAN on the CIFAR-10 dataset.** The connection update frequency of the generator $\Delta T$ is set to be 1000. The initial $\gamma$ is set to 0.5 and we use a cosine annealing function $f_{\text{decay}}$ following RigL and ITOP.

### B.3 DYNAMIC ADJUST AND DST HYPERPARAMETERS FOR DDST

**R-DDST.** For R-DDST, only the generator is adjusted using DST while the discriminator is adjusted using dynamic adjust (DA). The DA bounds are chosen to be $[0.475, 0.525]$, $[0.45, 0.55]$, and $[0.45, 0.55]$ for SNGAN (CIFAR-10), SNGAN (STL-10) and BigGAN (CIFAR-10), respectively. $\Delta d$ is set to be 0.05, 0.025, 0.05 for SNGAN (CIFAR-10), SNGAN (STL-10) and BigGAN (CIFAR-10), respectively. The density of the discriminator is adjusted every 1000, 2000, and 5000 iterations for the three settings, respectively. Time-averaged BR over 1000 iterations is used as the indicator. We use the same setting used in Section B.2 for the generator.

**S-DDST.** For S-DDST, the discriminator is adjusted using DA in the first half of training, i.e., the first 50,000 iterations. In the second half of the training, the discriminator is adjusted using DST. The generator is only adjusted with DST. For the DA bounds, they are set as $[0.45, 0.55]$ and $[0.475, 0.525]$ for CIFAR-10 and STL-10 dataset, respectively. The density of the discriminator is adjusted every 2000 iterations for each dataset. The density of the generator is adjusted every 1000 iterations.

We compute BR for every iteration to visualize the BR evolution, whereas one should note that such computational cost can be greatly decreased if BR is computed every $\Delta T$ iterations.

## C ALGORITHMS

In this section, we present the detailed algorithms for both DA and S-DDST. We do not present the algorithm of R-DDST as it is a combination of DA and SDST.

Table 4: ResNet architecture for CIFAR-10.

| (a) Generator | (b) Discriminator |
|---|---|
| $z \in \mathbb{R}^{128} \sim \mathcal{N}(0, I)$ | image $x \in [-1, 1]^{32 \times 32 \times 3}$ |
| dense, $4 \times 4 \times 256$ | ResBlock down 128 |
| ResBlock up 256 | ResBlock down 128 |
| ResBlock up 256 | ResBlock down 128 |
| ResBlock up 256 | ResBlock down 128 |
| BN, ReLU, $3 \times 3$ conv, Tanh | ReLU 0.1 |
| | Global sum pooling |
| | dense $\rightarrow 1$ |

Table 5: ResNet architecture for STL-10.

| (a) Generator | (b) Discriminator |
|---|---|
| $z \in \mathbb{R}^{128} \sim \mathcal{N}(0, I)$ | image $x \in [-1, 1]^{48 \times 48 \times 3}$ |
| dense, $6 \times 6 \times 512$ | ResBlock down 64 |
| ResBlock up 256 | ResBlock down 128 |
| ResBlock up 128 | ResBlock down 256 |
| ResBlock up 64 | ResBlock down 512 |
| BN, ReLU, $3 \times 3$ conv, Tanh | ResBlock down 1024 |
| | ReLU 0.1 |
| | Global avg pooling |
| | dense $\rightarrow 1$ |

## C.1 DYNAMIC ADJUST ALGORITHM

We first present DA in Algorithm 1.

---

**Algorithm 1** Dynamic density adjust (DA) for the discriminator.

---

**Require:** Generator $G$, discriminator $D$, DA upper bound $B_+$ and lower bound $B_-$, DA interval $\Delta T_D$, density increment $\Delta d$, grow method $\mathcal{A}$, drop method $\mathcal{B}$, iteration $t$.
1: **if** $t \mod \Delta T_D == 0$ **then**
2:     Compute time-averaged BR with Equation 3
3:     **if** BR is greater or equal to $B_+$ **then**
4:         Increase the density of discriminator from $d_D$ to $d_D + \Delta d$ using given grow method $\mathcal{A}$.
5:     **else if** BR is less or equal to $B_-$ **then**
6:         Decrease the density of discriminator from $d_D$ to $d_D - \Delta d$ using given drop method $\mathcal{B}$.
7:     **end if**
8: **end if**

---

## C.2 STRICT DOUBLE DYNAMIC SPARSE TRAINING ALGORITHM

Details of S-DDST algorithm is presented in Algorithm 2.

---

**Algorithm 2** Strict double dynamic sparse training (S-DDST) for GANs.

---

**Require:** Generator $G$, discriminator $D$, total number of iterations $T$, number of training steps for discriminator in each iteration $N$, maximal density of discriminator $d_{\max}$.
1: **for** $t$ in $[1, \cdots, T]$ **do**
2:     **for** $n$ in $[1, \cdots, N]$ **do**
3:         Compute the loss of discriminator $\mathcal{L}_D(\boldsymbol{\theta}_D)$
4:         $\mathcal{L}_D(\boldsymbol{\theta}_D).backward()$
5:     **end for**
6:     Compute the loss of generator $\mathcal{L}_G(\boldsymbol{\theta}_G)$
7:     $\mathcal{L}_G(\boldsymbol{\theta}_G).backward()$
8:     **if** $t$ is less than $0.5 * T$ and current density of discriminator $d_D$ is less than $d_{\max}$ **then**
9:         Apply DA in Algorithm 1 to $D$
10:     **else**
11:         Apply DST to $D$
12:     **end if**
13:     Apply DST to $G$
14: **end for**

---

## D   DDST BALANCE RATIO EVOLUTION

In this section, we show that DDST methods are able to maintain a BR throughout training. We show the time evolution of BR and discriminator density for CIFAR-10 and STL-10 datasets.

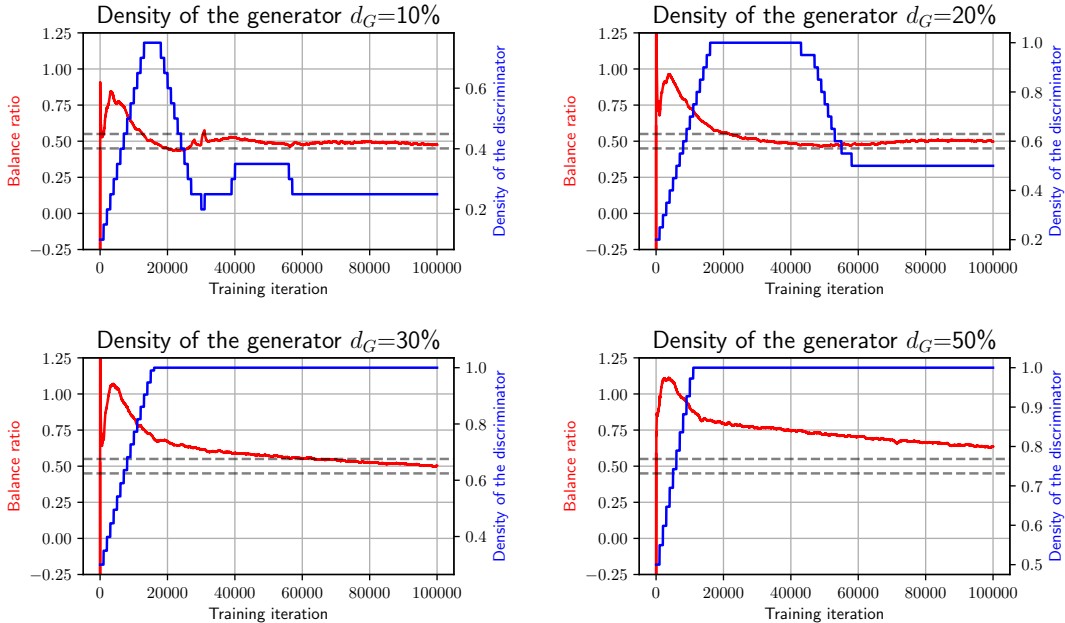

Figure 5: Balance ratio and discriminator density evolution during training for ● R-DDST(RigL) on CIFAR-10. Dashed lines represent BR values of 0.45 and 0.55.

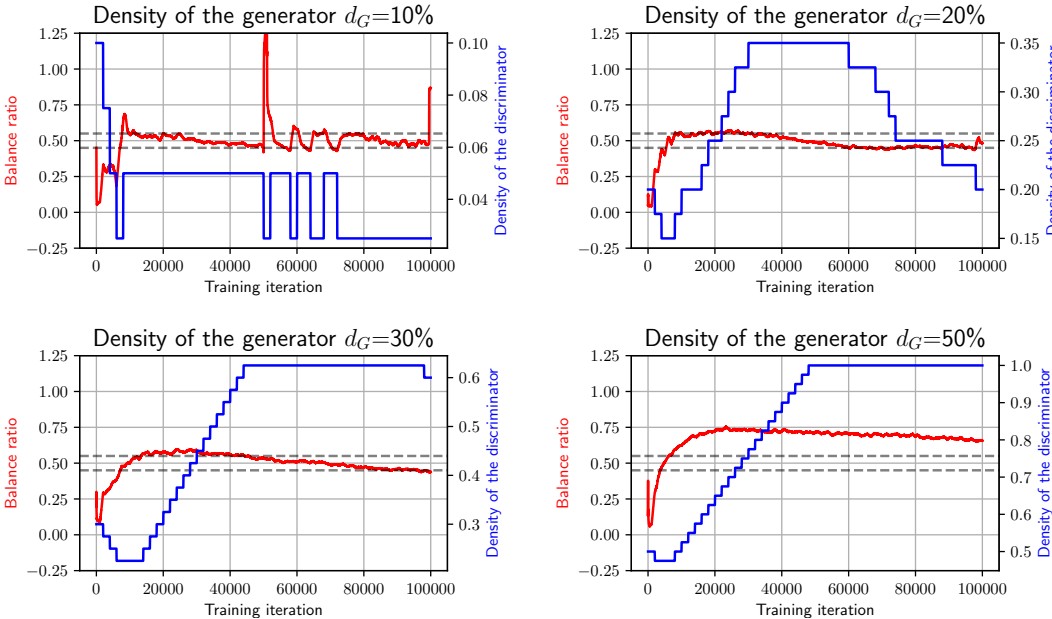

Figure 6: Balance ratio and discriminator density evolution during training for ● R-DDST(RigL) on STL-10. Dashed lines represent BR values of 0.45 and 0.55.

## D.1 R-DDST

Results are shown in Figure 5 and Figure 6. It clearly illustrates the ability of ● R-DDST(RigL) to keep the BR controlled during GAN training.

## D.2 S-DDST

Results of S-DDST are shown in Figure 7 and Figure 8. It clearly illustrates the ability of ● S-DDST(RigL) to keep the BR controlled during GAN training.

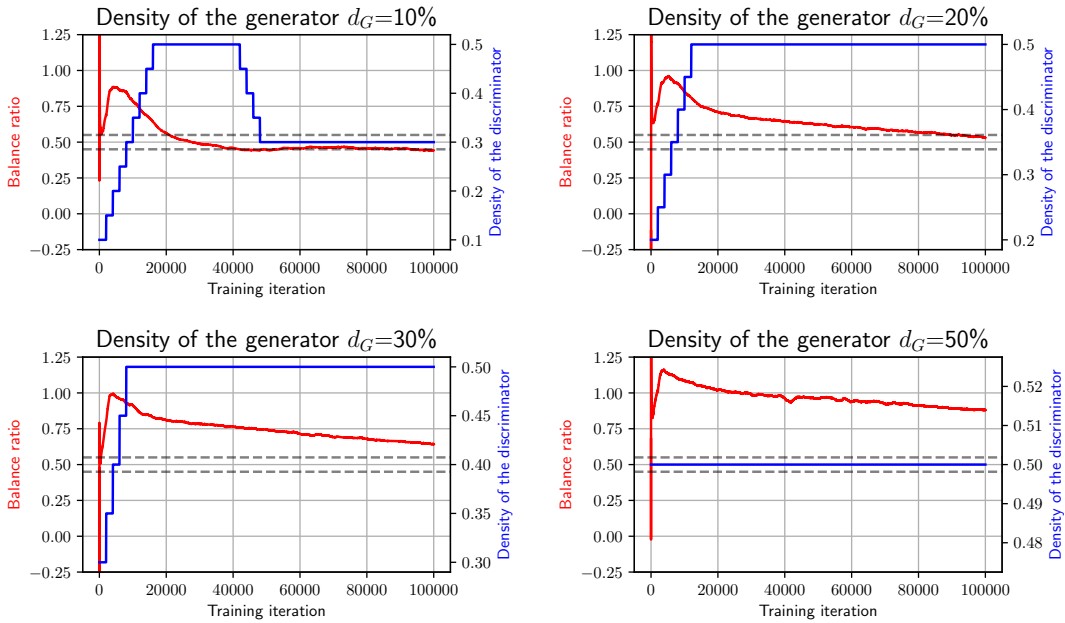

Figure 7: Balance ratio and discriminator density evolution during training for ● S-DDST(RigL) on CIFAR-10. Dashed lines represent BR values of 0.45 and 0.55.

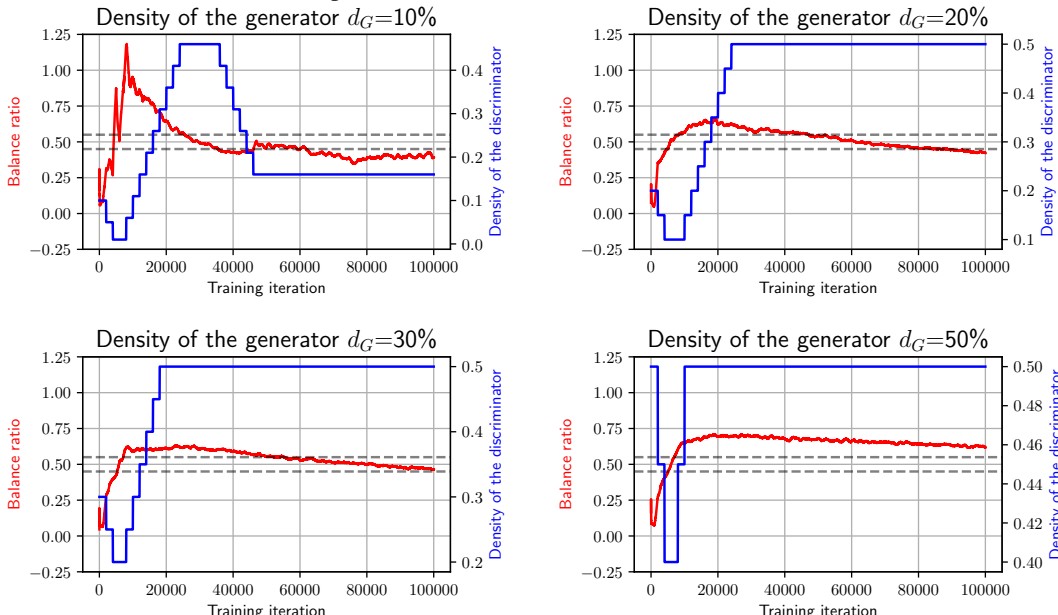

Figure 8: Balance ratio and discriminator density evolution during training for ● S-DDST(RigL) on STL-10. Dashed lines represent BR values of 0.45 and 0.55.

# E MORE EXPERIMENT RESULTS

In this section, we present IS scores results for Table 1 and Table 3. The corresponding results are shown in Table 6 and Table 7, respectively. We also include FID results of CIFAR-10 test set in Table 8.

Table 6: IS (higher is better) of different sparse training methods on CIFAR-10 and STL-10 datasets. There is no constraint on the density of the discriminator, i.e., $d_{max} = 100\%$.

| Dataset | CIFAR-10 | | | | STL-10 | | | |
|---|---|---|---|---|---|---|---|---|
| Generator density | 10% | 20 % | 30 % | 50 % | 10% | 20 % | 30 % | 50 % |
| (Dense Baseline) | 8.48 | | | | 9.16 | | | |
| Static-Balance | 7.18 | 7.76 | 8.01 | 8.31 | 7.84 | 8.07 | 8.35 | 8.60 |
| Static-Strong | 7.49 | 8.00 | 8.31 | 8.54 | 7.74 | 8.29 | 8.38 | 8.83 |
| ◆ SDST-Balance (SET) | 6.94 | 7.79 | 8.05 | 8.20 | 8.40 | 8.54 | 9.20 | 9.12 |
| ◆ SDST-Strong (SET) | 8.27 | 8.46 | 8.51 | 8.43 | 8.10 | 8.67 | 8.89 | 9.31 |
| ● SDST-Balance (RigL) | 6.81 | 7.77 | 8.08 | 8.30 | 8.85 | 8.74 | 9.19 | 9.14 |
| ● SDST-Strong (RigL) | 8.20 | 8.38 | 8.55 | 8.48 | 8.25 | 9.30 | 9.01 | 9.37 |
| ◆ R-DDST (SET) | 8.55 | 8.50 | 8.40 | 8.56 | 8.33 | 8.62 | 9.04 | 9.34 |
| ● R-DDST (RigL) | 8.32 | 8.61 | 8.49 | 8.55 | 8.79 | 9.25 | 9.30 | 9.27 |

Table 7: IS (higher is better) of different sparse training methods on CIFAR-10 and STL-10 datasets. The density of the discriminator is constrained to be lower than $d_{max} = 50\%$.

| Dataset | CIFAR-10 | | | | STL-10 | | | |
|---|---|---|---|---|---|---|---|---|
| Generator density | 10% | 20 % | 30 % | 50 % | 10% | 20 % | 30 % | 50 % |
| (Dense Baseline) | 8.48 | | | | 9.16 | | | |
| Static-Balance | 7.18 | 7.76 | 8.01 | 8.31 | 7.84 | 8.07 | 8.35 | 8.60 |
| Static-Strong | 7.86 | 8.21 | 8.35 | 8.28 | 7.81 | 8.05 | 8.26 | 8.37 |
| ◆ SDST-Balance (SET) | 6.94 | 7.79 | 8.05 | 8.20 | 8.40 | 8.54 | 9.20 | 9.12 |
| ◆ SDST-Strong (SET) | 8.22 | 8.36 | 8.56 | 8.35 | 8.40 | 9.29 | 9.21 | 9.22 |
| ● SDST-Balance (RigL) | 6.81 | 7.77 | 8.08 | 8.30 | 8.85 | 8.74 | 9.19 | 9.14 |
| ● SDST-Strong (RigL) | 8.24 | 8.51 | 8.20 | 8.18 | 7.70 | 9.32 | 9.19 | 9.33 |
| ◆ S-DDST (SET) | 8.08 | 8.25 | 8.45 | 8.23 | 8.07 | 8.91 | 9.11 | 9.50 |
| ● S-DDST (RigL) | 8.16 | 8.47 | 8.29 | 8.32 | 8.45 | 9.24 | 9.16 | 9.03 |

Table 8: FID of test set (↓) of different sparse training methods on CIFAR-10 dataset. Best results are in **bold**; second-best results are underlined.

| Maximal discriminator density $d_{max}$ | 100 % | | | | 50 % | | | |
|---|---|---|---|---|---|---|---|---|
| Generator density | 10% | 20 % | 30 % | 50 % | 10% | 20 % | 30 % | 50 % |
| (Dense Baseline) | 13.32 | | | | | | | |
| Static-Balance | 29.53 | 20.83 | 17.09 | 14.21 | 29.53 | 20.83 | 17.09 | 14.21 |
| Static-Strong | 29.15 | 22.17 | 17.37 | 14.04 | 21.98 | 19.35 | 16.52 | 14.84 |
| ◆ SDST-Balance (SET) | 30.34 | 21.00 | 16.84 | 15.53 | 30.34 | 21.00 | 16.84 | 15.53 |
| ◆ SDST-Strong (SET) | 19.95 | 17.05 | 15.16 | 14.10 | 18.83 | 15.96 | 15.61 | **14.53** |
| ● SDST-Balance (RigL) | 33.25 | 20.74 | 17.78 | 14.75 | 33.25 | 20.74 | 17.78 | 14.75 |
| ● SDST-Strong (RigL) | 19.67 | 15.79 | **13.89** | 14.36 | 18.60 | 16.01 | 15.84 | 14.67 |
| ◆ R-DDST (SET) | **16.34** | 15.29 | 14.30 | **13.85** | - | - | - | - |
| ● R-DDST (RigL) | 16.65 | **14.87** | 14.49 | 14.05 | - | - | - | - |
| ◆ S-DDST (SET) | - | - | - | - | 17.75 | 15.74 | 15.07 | 14.91 |
| ● S-DDST (RigL) | - | - | - | - | **17.07** | **15.50** | **15.02** | 14.67 |

Table 9: Training FLOPs ($\times 10^{17}$) of different sparse training methods on CIFAR-10 dataset.

| Dataset | | **CIFAR-10** | | | |
|---|---|---|---|---|---|
| Generator density | | 10% | 20 % | 30 % | 50 % |
| (Dense Baseline) | | (1.74, 1.00×) | | | |
| $d_{\max} = 100\%$ | Static-Strong | (0.63, 0.36×) | (0.70, 0.40×) | (0.80, 0.46×) | (1.07, 0.61×) |
| | SDST-Strong | (0.63, 0.36×) | (0.70, 0.40×) | (0.80, 0.46×) | (1.07, 0.61×) |
| | R-DDST | (0.63, 0.36×) | (0.70, 0.40×) | (0.80, 0.46×) | (1.07, 0.61×) |
| $d_{\max} = 50\%$ | Static-Strong | (0.36, 0.21×) | (0.43, 0.25×) | (0.53, 0.30×) | (0.79, 0.46×) |
| | SDST-Strong | (0.36, 0.21×) | (0.43, 0.25×) | (0.53, 0.30×) | (0.79, 0.46×) |
| | S-DDST | (0.36, 0.21×) | (0.43, 0.25×) | (0.53, 0.30×) | (0.79, 0.46×) |

Table 10: Training FLOPs ($\times 10^{17}$) of different sparse training methods on STL-10 dataset.

| Dataset | | **STL-10** | | | |
|---|---|---|---|---|---|
| Generator density | | 10% | 20 % | 30 % | 50 % |
| (Dense Baseline) | | (1.85, 1.00×) | | | |
| $d_{\max} = 100\%$ | Static-Strong | (1.30, 0.75×) | (1.34, 0.77×) | (1.36, 0.78×) | (1.41, 0.81×) |
| | SDST-Strong | (1.30, 0.75×) | (1.34, 0.77×) | (1.36, 0.78×) | (1.41, 0.81×) |
| | R-DDST | (1.30, 0.75×) | (1.34, 0.77×) | (1.36, 0.78×) | (1.41, 0.81×) |
| $d_{\max} = 50\%$ | Static-Strong | (1.07, 0.62×) | (1.11, 0.63×) | (1.13, 0.65×) | (1.18, 0.68×) |
| | SDST-Strong | (1.07, 0.62×) | (1.11, 0.63×) | (1.13, 0.65×) | (1.18, 0.68×) |
| | S-DDST | (1.07, 0.62×) | (1.11, 0.63×) | (1.13, 0.65×) | (1.18, 0.68×) |

## F  A ROUGH ESTIMATION OF COMPUTATIONAL COSTS ON SNGAN

In this section, we provide a very rough estimation on the computational cost of different sparse training methods in terms of training FLOPs. Please see Appendix G for a more accurate comparison. We approximate the number of backward FLOPs with two times the number of forward FLOPs. We compare the following methods under two settings where $d_{\max} \in \{100\%, 50\%\}$:

- Dense training.
- static-Strong.
- SDST-Strong.
- R-DDST.
- S-DDST.

We choose static-Strong and SDST-Strong as they perform relatively better than their counterparts with the balance strategy. **To simplify our calculation, we compute the FLOPs of R-DDST and S-DDST assuming the discriminator density $d_D = d_{\max}$.** We also assume that DST may not cause the change of FLOPs. The results are shown in Table 9 and Table 10.

It can be seen that the extra computational cost introduced by DA[3], which computes BR, and ● RigL, which computes gradient magnitude for connection growth, is negligible compared to the total training cost as they only happen every several hundred iterations.

## G  A DETAILED COMPARISON OF TRAINING COSTS

In this section, we compute the computational cost of ● RigL vairants and static baselines more accurately. More specifically, we take into account the density redistribution over different layers in this section. Also, we neglect the computational overhead introduced by computing BR.

---

[3]In our experiment, we compute BR for every iteration to visualize its evolution. However, BR only needs to be calculated for every several hundred iterations to compute the time-averaged BR.

Table 11: Training FLOPs ($\times 10^{17}$) and normalized training FLOPs with respect to dense training of different sparse training methods with SNGAN on CIFAR-10 dataset.

| Dataset | CIFAR-10 | | | |
|---|---|---|---|---|
| Generator density | 10% | 20 % | 30 % | 50 % |
| (Dense Baseline) | (2.67, 100%) | | | |
| Static-Balance | (0.24, 9.00%) | (0.46, 17.08%) | (0.70, 26.29% ) | (1.26, 47.34%) |
| Static-Strong | (1.56, 58.29%) | (1.63, 60.94%) | (1.72, 64.57%) | (1.99, 74.64%) |
| ● SDST-Balance (RigL) | (0.24, 9.14%) | (0.46, 17.19%) | (0.68, 25.62%) | (1.16, 43.45%) |
| ● SDST-Strong (RigL) | (1.57, 58.80%) | (1.64, 61.42%) | (1.71, 64.04%) | (1.90, 71.16%) |
| ● R-DDST (RigL) | (0.49, 18.20%) | (1.14, 42.73%) | (1.63, 60.94%) | (1.85, 69.33%) |

Table 12: Training FLOPs ($\times 10^{17}$) and normalized training FLOPs with respect to dense training of different sparse training methods with BigGAN on CIFAR-10 dataset.

| Dataset | CIFAR-10 | | | |
|---|---|---|---|---|
| Generator density | 10% | 20 % | 30 % | 50 % |
| (Dense Baseline) | (6.80, 100%) | | | |
| Static-Balance | (0.67, 9.78%) | (1.30, 19.04%) | (1.95, 28.69%) | (3.34, 49.09% ) |
| Static-Strong | (5.71, 83.90%) | (5.78, 83.90%) | (5.87, 86.34%) | (6.14, 90.26%) |
| ● SDST-Balance (RigL) | (0.67, 9.91%) | (1.32, 19.41%) | (1.96, 28.82%) | (3.29, 48.38%) |
| ● SDST-Strong (RigL) | (5.72, 84.04%) | (5.80, 85.22%) | (5.89, 86.54%) | (6.09, 89.56%) |
| ● R-DDST (RigL) | (0.67, 9.77%) | (1.69, 24.85%) | (2.72, 40.00%) | (5.25, 77.13%) |

## G.1 SNGAN ON THE CIFAR-10 DATASET

We first show the results of SNGAN (CIFAR-10) in Table 11. Combined with the results shown in Table 1, it shows that generally ● R-DDST is able to achieve promising performance with reasonable computational costs. More precisely, R-DDST outperforms ● SDST-Strong (RigL) with much fewer training FLOPs. The reason is that ● SDST-Strong (RigL) uses unnecessarily strong (dense) discriminators.

## G.2 BIGGAN ON THE CIFAR-10 DATASET

In this subsection, we show the results of BigGAN (CIFAR-10). We have included the simplified version in the Table 2. Here we give more detailed results in Table 12. The results are similar to SNGAN on the CIFAR dataset.

## H ONE-SHOT PRUNING AFTER TRAINING WITHOUT FINE-TUNING

In this section, we perform one-shot pruning after training for GANs without any fine-tuning. The results of SNGANs on the CIFAR-10 and STL-10 datasets are shown in Table 13.

Table 13: FID (↓) of different sparse training methods on CIFAR-10 and STL-10 datasets. The density of the discriminator is constrained to be lower than 50%. Best results are in **bold**; second-best results are underlined.

| | CIFAR-10 | | | | STL-10 | | | |
|---|---|---|---|---|---|---|---|---|
| Generator density | 10% | 20 % | 30 % | 50 % | 10% | 20 % | 30 % | 50 % |
| (Dense Baseline) | 10.74 | | | | 29.71 | | | |
| PF without fine-tuning | 305.81 | 247.99 | 89.29 | 30.03 | 339.95 | 195.69 | 156.29 | 66.66 |
| Static-balance | 26.73 | 18.04 | 14.38 | 12.22 | 50.08 | 44.19 | 43.96 | 37.21 |
| Static-strong | 22.35 | 16.57 | 13.47 | 12.22 | 50.28 | 44.95 | 42.12 | 37.21 |
| ◆ DST-balance (SET) | 32.02 | 18.54 | 14.74 | 13.23 | 49.91 | 33.71 | 32.92 | 31.75 |
| ● DST-balance (RigL) | 24.56 | 15.53 | 13.62 | 12.51 | 66.90 | 50.34 | 44.57 | 32.63 |
| ◆ SDST-balance (SET) | 27.80 | 18.13 | 14.15 | 12.32 | 63.57 | 49.05 | 43.74 | 31.29 |
| ◆ SDST-strong (SET) | 16.00 | 13.31 | 13.17 | 12.32 | 48.40 | 33.56 | 32.19 | 31.29 |
| ● SDST-balance (RigL) | 30.38 | 17.89 | 14.95 | 12.09 | 46.17 | 38.12 | 32.48 | 31.30 |
| ● SDST-strong (RigL) | 15.66 | 13.20 | 12.99 | 12.09 | 63.65 | 33.45 | 32.09 | 31.30 |
| ◆ S-DDST (SET) | 14.22 | 13.30 | 12.39 | **11.97** | 51.72 | 35.74 | 42.36 | 31.68 |
| ● S-DDST (RigL) | **14.13** | **12.87** | **12.15** | 12.17 | **44.28** | **32.84** | **32.00** | **30.28** |

