# OpenReview forum: "Double dynamic sparse training for GANs"
_ICLR.cc/2023/Conference — Submitted to ICLR 2023_

### Official Review · Reviewer_edZK · 2022-10-20

**Confidence:** 4
**Correctness:** 3
**Technical Novelty And Significance:** 2
**Empirical Novelty And Significance:** 1
**Recommendation:** 3

**Clarity, Quality, Novelty And Reproducibility:**

Clarity is good.

Quality and Novelty are low.

Reproducibility is fine.

**Strength And Weaknesses:**

Pros.

1. The writing is good and easy to follow.
2. The design of the balance ratio makes sense.



Cons.

1. Lack of insights and incremental novelty. The balance problem has already been pointed out in previous GAN pruning papers. The authors directly apply dynamic sparse training to the GAN model. Furthermore, directly training sparse GANs has also been explored by (Liu et al., 2022).
2. The balance ratio also has design limitations since it is sparsity/density invariant. It can not directly reflect the capacity balance or topology balance between the generator and discriminator.
3. It is very hard to read the results in Fig 3.
4. The paper claims efficient GAN. However, only small GANs like SNGAN are evaluated on small datasets like CIFAR-10 and STL-10. Large datasets with advanced GANs are highly needed, like ImageNet/FFHQ with BigGAN and StyleGAN.

**Summary Of The Paper:**

Summary.

This paper is dedicated to developing efficient generative adversarial networks (GANs). The authors apply dynamic sparsity training to GAN models. They point out that the balance of the generator and discriminator is the key to reaching good performance. The authors propose a quantity BR to measure the balance and use dynamic sparsity training for both the generator and discriminator. STL-10 and CIFAR-10 are used for their experiments.

**Summary Of The Review:**

Limited novelty and highly insufficient experiments.

---

> ### Author Response · Authors · 2022-11-19
> **Response to reviewer edZK [Part 2/2]**
>
> > The balance ratio also has design limitations since it is sparsity/density invariant. It can not directly reflect the capacity balance or topology balance between the generator and discriminator.
>
> `Answer 2` In Section 3.3, we have shown that the balance ratio can indeed reflect the unbalance between the sparse generator and the sparse discriminator when the dense backbones (or maximum capacity) are fixed. The balance ratio works because, for example, a relatively larger discriminator would learn faster (with stronger representation and optimization power) compared to an unbalanced smaller generator, leading to a smaller BR. So we assume that the "sparsity/density invariant" mentioned by the reviewer means that BR is unable to tell how sparse a generator/discriminator is if we have no knowledge of the dense backbones. If that is the case, we agree with the reviewer that BR is "sparsity/density invariant". We would like to answer the reviewer's concern from the following two perspectives:
>
> 1. Our intention is exactly finding a "sparsity/density invariant" indicator. We would like to give an easy example here. For example, suppose that we can find two discriminators that nearly present the same function class and with similar optimization/generalization power, where one of them is large but sparse and the other one is small but dense. Under such circumstances, we believe that the two discriminators should behave similarly in a given GAN with the same generator. As a consequence, BR should give the same results and hence "sparsity/density invariant".
>
> 2. Moreover, we respectfully disagree with the reviewer that this is a design limitation. On the contrary, we believe that BR can be easily computed and hence can be possibly applied to many different architectures and settings.
>
>
> > It is very hard to read the results in Fig 3.
>
> `Answer 3` We thank the reviewer for the feedback. The reason that Fig 3 is hard to read is that we have included cases where training failure happens, i.e. when the discriminator is not strong enough so that it does not provide useful information. As a consequence, BR wildly oscillates. The fact that we want to show BR is able to reflect such training failures makes Fig.3 inherently difficult to interpret. Currently, we cannot think of a good way to improve the plot. Any suggestions would be greatly appreciated.
>
> > The paper claims efficient GAN. However, only small GANs like SNGAN are evaluated on small datasets like CIFAR-10 and STL-10. Large datasets with advanced GANs are highly needed, like ImageNet/FFHQ with BigGAN and StyleGAN.
>
> `Answer 4` We thank the reviewer for the valuable comment. In our updated manuscript, we have included BigGAN on CIFAR-10 experiments in Table 2 due to time and computational resource limitations. We show that R-DDST can outperform the strong baseline SDST-Strong (RigL) with much less training cost. Please also refer to Section 5.1 for more details.
>
> [1] Yu, Chong, and Jeff Pool. "Self-supervised generative adversarial compression." Advances in Neural Information Processing Systems 33 (2020): 8235-8246.
>
> [2] Li, Shaojie, et al. "Revisiting discriminator in GAN compression: A generator-discriminator cooperative compression scheme." Advances in Neural Information Processing Systems 34 (2021): 28560-28572.
>
> [3] Liu, Shiwei, et al. "Don't Be So Dense: Sparse-to-Sparse GAN Training Without Sacrificing Performance." arXiv preprint arXiv:2203.02770 (2022).

---

> ### Author Response · Authors · 2022-11-19
> **Response to reviewer edZK [Part 1/2]**
>
> We thank the reviewer for acknowledging that our work is clear and the introduced BR makes sense. We address the reviewer's concerns as follows:
>
> > Lack of insights and incremental novelty. The balance problem has already been pointed out in previous GAN pruning papers. The authors directly apply dynamic sparse training to the GAN model. Furthermore, directly training sparse GANs has also been explored by (Liu et al., 2022).
>
> `Answer 1` We thank the reviewer's comment. We would like to clarify our contribution and novelty here:
>
> - Indeed, many works have pointed out that balance is important in GAN pruning or compression [1,2,3]. To the best of our knowledge, GCC [2] is the only GAN compression work that proposes a metric to quantify the balance of the generator and the discriminator. However, this metric cannot be directly applied to GAN dynamic sparse training as the metric calculation requires a teacher generator, which is a pre-trained dense network. Hence, we believe coming up with a new and efficient indicator for GAN DST is indeed novel.
>
> - Moreover, we want to kindly point out that we are not directly applying DST to the GAN models. We would like to give a comparison between our DDST and STU-GAN [3] (or SDST in general), which is directly applying DST to GANs with some interesting tricks and is almost identical to SDST (RigL), here:
>
>   - **Comparison between R-DDST (RigL) and STU-GAN.** The main difference between the two is whether the density of the discriminator is fixed during training. In STU-GAN, the discriminator density is chosen manually at initialization and fixed thereafter throughout training. The problem is that it is not guaranteed that the manually chosen discriminator density is a good choice. Additionally, it is not practical to tune manually in real-world scenarios as it does not align with the intention of DST to reduce computational costs.
>
>     On the contrary, for R-DDST, the density of the discriminator will automatically be adjusted with BR as the indicator. More precisely, if BR shows that a stronger discriminator is needed, the density of the discriminator will increase, and vice versa. The flexibility in the discriminator density ensures that (1) we no longer need to choose discriminator density manually at initialization; (2) we do not need to waste computational cost for an overly-strong discriminator which leads to even worse performance. To see this, please check our new experiments regarding FID and training FLOPs for BigGAN on CIFAR-10 in Table 2. Our R-DDST shows better performance than SDST-Strong (RigL) with much less computational costs.
>
>   - **Comparison between S-DDST (RigL) and STU-GAN.** For our second method, S-DDST, there exist two phases. More specifically, during the first phase, S-DDST will be just like R-DDST to find a suitable discriminator density $d_D^*$. Once the first phase ends, the density will be fixed for the following second phase. And the discriminator will perform DST during the second phase, i.e., grow and drop connections with density fixed just like the generator. Notice that in both R-DDST and STU-GAN (or SDST in general), the discriminator does not perform DST, which means they do not fully utilize the in-time-over-parameterization advantage provided by the sparse training on the discriminator side.
>
>     We have also updated our manuscript to better reflect the difference between R-DDST, S-DDST, and STU-GAN (or SDST in general) in the introduction, related works, Section 4, and Section 5.
>
> However, we agree with the reviewer that the importance of balance in GAN has been explored in the previous works so we removed the first bullet point in our contribution.

---

### Official Review · Reviewer_J5kM · 2022-10-23

**Confidence:** 4
**Correctness:** 3
**Technical Novelty And Significance:** 2
**Empirical Novelty And Significance:** 3
**Recommendation:** 3

**Clarity, Quality, Novelty And Reproducibility:**

This paper is well written. However, more experiments should be conducted to verify the effectiveness of the proposed approach.

**Strength And Weaknesses:**

The authors propose a novel metric to balance the dynamic sparse training technique in generative neural networks. The problem is challenging and the experiments are extensive. However, before the reviewer recommends the acceptance of this manuscript, several concerns should be solved:
(i)	Applying the DST technique for training GAN is not new. Liu et al 2022 also propose STU-GAN to train GANs. We recommend the authors give a more detailed discussion with STU-GAN in the introduction and related work section. The current discussion with STU-GAN needs further clarification.
(ii)	The results in the experiments merely restrict to SNGAN, which is too specific to conclude the phenomenon. We recommend the authors provide more exploration on other typical unconditional GANs, such as BIGAN (CIFAR10 or ImageNet with low resolution) and StyleGAN2 (CIFAR10 or FFHQ).
(iii)	No baselines are provided in the experiments. We recommend the authors compare their results with several unconditional GAN baselines. The current experiments cannot tell the readers which sparse training technique is better for obtaining a lightweight GAN.
(iv)	The unstructured sparse GAN training can not achieve real training acceleration or inference acceleration.  We recommend the authors extend the weight dynamic sparsity to channel dynamic sparsity. In addition, more baselines are required to demonstrate the effectiveness of the proposed approach.
(v)	Can the author explain why the performance is largely degraded compared with dense training? In general, applying dynamic sparse training techniques on training ResNet, there is almost no performance degradation.


**Summary Of The Paper:**

In this work, the authors extend the dynamic sparse training (DST) technique to train unconditional GANs(SNGAN). In addition, the authors propose a new metric called balance ratio to explore the role of sparsity in generator and discriminator. Extensive experiments are conducted to show the influence of different sparsity of generators and discriminators in SNGAN.

**Summary Of The Review:**

See the comments above.

---

> ### Author Response · Authors · 2022-11-19
> **Response to reviewer J5kM [Part 2/2]**
>
> > The unstructured sparse GAN training can not achieve real training acceleration or inference acceleration. We recommend the authors extend the weight dynamic sparsity to channel dynamic sparsity.
>
> `Answer 4` We totally agree with the reviewer that DST with structured pruning will bring much better real-world benefits compared to unstructured pruning. We would like to answer the reviewer's questions from two perspectives:
>
> 1. Based on the fact that the works in the GAN DST research field are very limited, we believe that currently investigating unstructured pruning is still valuable to the field. We will leave the exploration of structured pruning to future research.
>
> 2. Part of our contribution is about investigating the unbalance in sparse GAN training. We believe that our work is not purely application-oriented. More specifically, our work tries to address the unbalance problem in sparse GAN training and attempts to potentially give some more insights into the sparse GAN training field.
>
> > Can the author explain why the performance is largely degraded compared with dense training? In general, applying dynamic sparse training techniques on training ResNet, there is almost no performance degradation.
>
> `Answer 5` We thank the reviewer for raising such an interesting question. We believe the reviewer is referring to the difference between pruning in image generation and image classification settings. We would like to elaborate on the difference below:
>
> 1. For models used in image classification tasks, feature dimensions shrink and finally become a 1D vector as the network deepens. It means that the output perturbation introduced by removing some of the weights may be mitigated by the pooling/downsample operations. That is possibly why in these classification models, we can trim down a large portion of the weights without greatly affecting the model performance.
>
> 2. However, for image generation models like GANs, the generator has many upsampling layers and the feature dimensions increase and finally become a 3D image. In this different setting, the output perturbation introduced by removing weights will be augmented during the forward pass of the neural network.
>
> We have also included one-shot pruning after training without fine-tuning results in Appendix H. It can be clearly seen that removing weights in a one-shot manner after training induces much more severe performance degradation compared to a supervised learning setting. This again shows the difference between the two kinds of models and the importance of DST methods in the field of pruning GANs.

---

> ### Author Response · Authors · 2022-11-19
> **Response to reviewer J5kM [Part 1/2]**
>
> We thank the reviewer for acknowledging that our work is novel and well-motivated. Below please find the responses to your questions and we sincerely hope they could address your concerns:
>
> > Applying the DST technique for training GAN is not new. Liu et al 2022 also propose STU-GAN to train GANs. We recommend the authors give a more detailed discussion with STU-GAN in the introduction and related work section. The current discussion with STU-GAN needs further clarification.
>
> `Answer 1` We thank the reviewer for the constructive suggestion. We have modified our manuscript to include discussions about the difference between DDST, SDST, and STU-GAN. More specifically, we have added related sentences, and paragraphs in the introduction, related work, Section 4, and Section 5 to help the readers to have a better understanding of the difference between these methods.
>
> > The results in the experiments merely restrict to SNGAN, which is too specific to conclude the phenomenon. We recommend the authors provide more exploration on other typical unconditional GANs, such as BIGAN (CIFAR10 or ImageNet with low resolution) and StyleGAN2 (CIFAR10 or FFHQ).
>
> `Answer 2` Thank you for your comment. We have included the BigGAN on CIFAR-10 experiments in Table 2 of our updated manuscript along with the required training FLOPs. The results show that our proposed R-DDST method can outperform the strong baseline SDST-strong (RigL) with much less computational cost.
>
> > No baselines are provided in the experiments. We recommend the authors compare their results with several unconditional GAN baselines. The current experiments cannot tell the readers which sparse training technique is better for obtaining a lightweight GAN. In addition, more baselines are required to demonstrate the effectiveness of the proposed approach.
>
> `Answer 3` We thank the reviewer for the valuable comment. If we understand the reviewer's questions correctly, the reviewer is referring to pruning baselines like static sparse training, GAN tickets, and fine-tuning after pruning (PF) adopted in the STU-GAN paper. We would like to answer the reviewer's questions as follows:
>
> 1. We believe that only static sparse training is a sparse training method while the other two are not. In fact, we have indeed included the static sparse training in our comparison and it shows that for almost every case DST variants are better than static sparse training methods.
>
> 2. As reported by STU-GAN, STU-GAN outperforms these baseline methods including static sparse training, GAN tickets, and fine-tuning after training. In our experiments, we compare our methods with SDST-Strong (RigL) and SDST-balance (RigL), which are practical versions (since STU-GAN needs a manual choice of the discriminator density) of STU-GAN. It should be able to show the value of DDST methods.
>
> 3. Though we did not actually implement the detailed comparison with these strong baselines as STU-GAN did, we have included one more baseline as the reviewer suggested. We have added one-shot pruning after training without fine-tuning as a baseline in Appendix H Table 13. It shows that directly pruning GANs in a one-shot manner induces a significant performance drop to the fully-trained GANs, this further shows the effectiveness and importance of DST methods.

---

### Official Review · Reviewer_cSFY · 2022-10-24

**Confidence:** 4
**Correctness:** 3
**Technical Novelty And Significance:** 3
**Empirical Novelty And Significance:** 2
**Recommendation:** 5

**Clarity, Quality, Novelty And Reproducibility:**

The paper is well-motivated. The method is sound but with insufficient evaluation and marginal performance gains.

**Strength And Weaknesses:**

I will first go through the strength of the paper:
## Strength
(1) The paper is well-motivated. STU-GAN only dynamically changes the sparsity of the generator, leaving the discriminator fixed. The proposed double dynamic sparsity can balance the sparsity of GAN's two components during training, which plays a similar role as the famous Stylegan2-ADA in dense GAN training (https://arxiv.org/pdf/2006.06676.pdf).

(2) The capacity of balance ratio to reflect the overfitting problem of sparse GANs is interesting.

(3) The performance improvement over SGAN demonstrate the effectiveness of DDST.

## Weakness
(1) While the proposed method is sound, the evaluation is insufficient. DDST is solely evaluated with SNGAN on small datasets such as cifar10 and STL-10. To draw solid conclusions on the practical usage, the evaluation of DDST on large-scale settings is required, such as the de-facto standards for conditional image generation BigGAN on ImageNet, or StyleGAN2 on FFQH.

(2) The method is relatively complicated (many hyperparameters and dynamically adjusted sparsity levels), while the performance improvement over the baseline SDST-strong (RigL) is marginal. I have a concern that the method is too complicated and not practical.

(3) While DDST shows better performance than STU-GAN, it uses relatively larger discriminators (around 50% sparsity), which intuitively increases the training FLOPs. Given the marginal performance improvement, the contribution of DDST is diluted. Moreover, why the training FLOPs of DDST and SDST are exactly the same? I expect DDST to have higher training FLOPs than SDST. Coming up with some smart ways to adjust the capacity of discriminators without changing sparsity would be more convincing, like playing with random pruning or adding some perturbations when overfitting happens.

(4) The balance ratio is very similar to the overfitting heuristics used in StyleGAN2-ADA (https://arxiv.org/pdf/2006.06676.pdf), but without clearly citing it. Is balance ratio a better heuristic than ADA?

(5) I am confused by the second meaning of the word double: the discriminator enjoys two levels of dynamic flexibility, namely density level and parameter level.



**Summary Of The Paper:**

This paper proposed double dynamic sparse training to train sparse GANs from scratch dotted DDST. It directly follows the work of STU-GAN (Liu2022) (a sparse GAN training approach only dynamically change the topology of generators) and proposes balance ratio to further adjust the sparsity of discriminators during training. They first confirm the finding of STU-GAN that the balance between discriminator and generator is important for sparse GAN training and then observe stronger discriminators usually lead to better GAN performance. The proposed balance ratio is a good indicator to check the overfitting problem of discriminators and further adjust the sparsity. The results on the small dataset CIFAR-10 and STL-10 demonstrate the effectiveness of DDST.

**Summary Of The Review:**

Overall, I like the idea of dynamic adjusting both discriminators and generators. However, the experiment is not very solid and the method itself is too complicated compared with the marginal performance gains.

---

> ### Author Response · Authors · 2022-11-19
> **Response to reviewer cSFY [Part 3/3]**
>
> > The balance ratio is very similar to the overfitting heuristics used in StyleGAN2-ADA (https://arxiv.org/pdf/2006.06676.pdf), but without clearly citing it. Is balance ratio a better heuristic than ADA?
>
> `Answer 4` We thank the reviewer for pointing out the similarity between StyleGAN2-ADA and DDST. We would like to clarify the differences and similarities here:
>
> 1. The first difference is that ADA tries to measure the overfitting of the discriminator to the **training data** for a limited-data case while BR tries to quantify the relative balance of the generator and the discriminator.
>
> 2. $r_v$ defined in ADA needs two separate dataset splits (training and validation set) to calculate the value while BR only needs one single dataset.
>
> 3. Admittedly, ADA and DDST indeed share the similarity that they both adjust a quantity (augmentation probability p for StyleGAN2-ADA and $d_D$ for R-DDST) with a criterion based on the discriminator output ($r_v$ for StyleGAN2-ADA and BR for R-DDST).
>
> We agree that DDST is in spirit very similar to ADA. However, we do not intend to compare BR to $r_v$ in ADA since we believe they have different purposes. We thank again the reviewer for pointing this out and we have updated Section 3.4 in our manuscript to reflect the similarity between the two and included the reference.
>
> > I am confused by the second meaning of the word double: the discriminator enjoys two levels of dynamic flexibility, namely density level and parameter level.
>
> `Answer 5` For this paragraph, we are actually referring to S-DDST, which is introduced in Section 5.2. We would like to answer the reviewer's question by explaining the two phases of S-DDST. More specifically, the two phases of DDST work as follows:
>
> 1. **Phase 1: density exploration.** During the first phase, S-DDST will perform DA algorithm just like R-DDST, with the exception that the maximum allowable discriminator density is strictly less than 100%. During this phase, S-DDST aims to find a suitable discriminator density $d_D^*$ with BR as the indicator. Hence, S-DDST enjoys density-level flexibility.
>
> 2. **Phase 2: parameter exploration.** During the second phase of S-DDST, the discriminator will fix the found discriminator density $d_D^*$ in the first phase and perform classical DST, i.e., the discriminator will grow and drop connections with density fixed. In this sense, S-DDST enjoys parameter-level flexibility.
>
>
> We have also added a paragraph in Section 5.3 to clarify the two levels of dynamic flexibility of S-DDST.

---

> ### Author Response · Authors · 2022-11-19
> **Response to reviewer cSFY [Part 2/3]**
>
> > While DDST shows better performance than STU-GAN, it uses relatively larger discriminators (around 50% sparsity), which intuitively increases the training FLOPs. Given the marginal performance improvement, the contribution of DDST is diluted. Moreover, why the training FLOPs of DDST and SDST are exactly the same? I expect DDST to have higher training FLOPs than SDST. Coming up with some smart ways to adjust the capacity of discriminators without changing sparsity would be more convincing, like playing with random pruning or adding some perturbations when overfitting happens.
>
> `Answer 3` We believe that there might be misunderstandings regarding the difference between STU-GAN and DDST. Firstly, we want to kindly point out that SDST is a generalization of STU-GAN. More specifically, STU-GAN is almost identical to SDST (RigL). To make the comparison between our work and STU-GAN clearer, we have added paragraphs in sections 4 and 5 to emphasize the difference.
>
> As for training FLOPs, we want to clarify as follows:
>
> 1. **For STU-GAN (or SDST in general), the discriminator density is manually chosen before the training and fixed thereafter.** For example, in STU-GAN, authors manually choose $d_D \in (10\\%, 30\\%, 50\\%, 100\\%)$. However, it is unclear how to choose the optimal $d_D$ when a target generator density $d_G$ is given. It is not practical to search for the optimal value as it does not align with the intention of sparse training which tries to reduce computational costs. Hence, in section 5 we provide two strategies to define the discriminator density for STU-GAN (or SDST in general), namely the strong strategy and the balance strategy.
>
> 2. **The discriminator density of DDST is indeed changing during training, which is the main difference compared to STU-GAN (SDST).** The discriminator density of DDST is dynamically adjusted during the training with BR as the indicator. More precisely, the discriminator density will increase if a stronger discriminator is needed, and vice versa. Hence, the training FLOPs of DDST are not necessarily larger/smaller than SDST. Based on our experiments, the training FLOPs of DDST are usually larger than SDST-Balance (RigL) and less than SDST-Strong (RigL). Please also see Appendix D for the balance ratio and discriminator density evolution during the training.
>
> 3. **Regarding why training FLOPs of DDST is exactly the same as SDST in Appendix F, we actually greatly simplify the calculation by assuming that DDST uses discriminator density $d_{max}$ throughout training.** So, Appendix F is a very rough estimation since generally the discriminator density of DDST is lower than $d_{max}$. During the rebuttal period, we implemented a more accurate FLOPs calculator to better visualize the advantage of computational cost savings achieved by DDST. More specifically, we have included the training FLOPs calculation in Table 2 for BigGAN on CIFAR-10. The results show that R-DDST performs better than SDST-Strong (RigL) with much less computational cost as SDST-Strong (RigL) always uses dense discriminators. The reviewer may also want to check results regarding training FLOPs for SNGAN on CIFAR-10 in Appendix G.
>
> To better explain the ideas of our proposed DDST, we have included more details with highlighted colors in sections 4 and 5. Please also see our `Answer 5` regarding how DDST works. We sincerely hope the updated manuscript along with `Answer 5` could explain the details of R-DDST and S-DDST better.

---

> ### Author Response · Authors · 2022-11-19
> **Response to reviewer cSFY [Part 1/3]**
>
> We thank the reviewer for acknowledging that our work is well-motivated and interesting. We hereby answer the reviewer's questions:
>
> > While the proposed method is sound, the evaluation is insufficient. DDST is solely evaluated with SNGAN on small datasets such as cifar10 and STL-10. To draw solid conclusions on the practical usage, the evaluation of DDST on large-scale settings is required, such as the de-facto standards for conditional image generation BigGAN on ImageNet, or StyleGAN2 on FFQH.
>
> `Answer 1` We thank the reviewer for the suggestion. We agree with the reviewer that a larger model or a larger dataset should be tested. Due to the time and computational resource limitations, we have added further experiments with BigGAN on the CIFAR-10 dataset. The results are shown in Table 2. The results show that R-DDST (RigL) is still able to achieve stable and superior performance compared to baselines most of the time. More specifically, R-DDST (RigL) is able to outperform SDST-strong (RigL) with much less computational cost.
>
> We are willing to include more results with larger models and datasets if time and computational resource permit.
>
> > The method is relatively complicated (many hyperparameters and dynamically adjusted sparsity levels), while the performance improvement over the baseline SDST-strong (RigL) is marginal. I have a concern that the method is too complicated and not practical.
>
> `Answer 2` We thank the reviewer for raising the concern that whether DDST can be practical. We want to address the reviewer's concerns from the following two perspectives:
>
> - **"The method is relatively complicated."** We want to kindly point out that our method mainly introduces 3 extra hyperparameters compared to STU-GAN (or SDST in general):
>
>   - the DA-bounds $[B_-,B_+]$, which is symmetric with respect to 0.5. In our experiments, we choose from $[0.45, 0.55]$ and $[0.475, 0.525]$.
>
>   - The pre-defined density increment $\Delta d$. In our experiments, we choose from 2.5% and 5%.
>
>   - The discriminator update frequency $\Delta T_D$. Notice that this hyperparameter is needed as long as one wants to perform DST on the discriminator.
>
> - **"The performance improvement over the baseline SDST-strong (RigL) is marginal."** We want to kindly point out that our method DDST does not only bring performance improvement but also significantly reduces computational costs. More specifically, we present detailed training FLOPs of different methods on BigGAN (CIFAR-10) in Table 2. It shows that SDST-strong (RigL) always has normalized training FLOPs larger than 80% as it always uses the dense discriminator. On the contrary, R-DDST has a better performance compared to SDST-strong (RigL) with much reduced computational cost. More results including training FLOPs for SNGAN on CIFAR-10 can be found in Appendix G.
>
>
> Put simply, we believe our method is not too complicated and it brings more than just the improvement of image quality.

---

### Official Review · Reviewer_Fvhb · 2022-10-25

**Confidence:** 5
**Clarity, Quality, Novelty And Reproducibility:** See the strength and weaknesses.
**Correctness:** 3
**Technical Novelty And Significance:** 2
**Empirical Novelty And Significance:** 2
**Recommendation:** 5

**Strength And Weaknesses:**

Strength:

1. The proposed quantity can be integrated with existing sparse training algorithms flexibly to stabilize the training process.

2. The experimental results show that the proposed method can stabilize the training process.

Weaknesses:

1. The performance in Table 2 shows that the improvements achieved by the proposed method over SDST are marginal in some tasks. The authors are recommended to look into this phenomenon.

2. Some theoretical analysis is needed for the proposed balance ratio. I understand it is difficult to prove the convergence of training.  But some analysis is essential to show the insights and properties of the proposed balance ratio, this paper looks too heuristic.

3. This paper is not well organized. For example, the authors present SDST for more than 1 page, which can potentially make the readers confused.

4. Some notations are abused. For example, the $\alpha$ in balance ratio and $f_{decay}$ should be different.


**Summary Of The Paper:**

In this paper, the authors propose a new dynamic sparse training method for GANs. The key idea is to balance the performance of the generator and discriminator during training by developing a quantity called balance ratio. Based on this quantity, the authors adjust the densities of the generator and discriminator during training to keep balance. This quantity can be integrated with existing sparse training algorithms flexibly. Some experiments are conducted to evaluate the performance of the proposed method.

**Summary Of The Review:**

The performance of the proposed method is marginal in some tasks.

This paper looks too heuristic since no theoretical analysis are provided for the proposed balance ratio.

This paper is also not well organized.

---

> ### Author Response · Authors · 2022-11-19
> **Response to reviewer Fvhb**
>
> We thank the reviewer for acknowledging that our proposed method is flexible. Below please find the responses to some specific comments and questions.
>
> > The performance in Table 2 shows that the improvements achieved by the proposed method over SDST are marginal in some tasks. The authors are recommended to look into this phenomenon.
>
> `Answer 1` We thank the reviewer for the helpful suggestion. We agree with the reviewer that in some cases the improvements achieved by our proposed DDST methods are not very significant. We would like to address the concern from the following two perspectives:
>
> 1. **Most mentioned cases appear at higher density ratios.** For these density ratios, the generator is still dense enough so that even static sparse training baselines are good enough. As a consequence, it is relatively hard to distinguish the performance difference between different methods. Hence, we believe that we should place more emphasis on performance improvement at lower density ratios.
>
> 2. **In a sparse training setting, we care about not only the quality of generated images but also computational costs.** For example, in almost all testing scenarios, SDST-Strong (RigL) is better than SDST-balance (RigL) but it requires much more computational resources as it always uses the largest possible discriminator. As a consequence, the large computational cost induced by the dense discriminator training can not be avoided for SDST-Strong (RigL).
>
>   To better visualize the benefits of R-DDST over SDST, we further conduct experiments on BigGAN (CIFAR-10). FID and detailed normalized training FLOPs are shown in Table 2. We see that the normalized training FLOPs of SDST-Strong (RigL) are always larger than 80% of the dense GAN training due to the computational cost of the dense discriminator. On the contrary, SDST-Balance (RigL) has much less computational cost while its performance is worse. However, R-DDST (RigL) shows improved performance compared to SDST-Strong (RigL) with much less computational cost, which means R-DDST achieves better test performance with much less computational cost.
>
>
> Put simply, DDST shows stable and superior performance in almost all testing cases. The advantage is most distinguishable for low density ratio cases. Moreover, DDST can find suitable discriminator density, which not only improves the performance of SDST but also avoid unnecessary extra computational overhead.
>
> > Some theoretical analysis is needed for the proposed balance ratio. I understand it is difficult to prove the convergence of training. But some analysis is essential to show the insights and properties of the proposed balance ratio, this paper looks too heuristic.
>
> `Answer 2` We are very thankful for the constructive suggestion. We agree with the reviewer that it is important to have a deeper understanding of sparse GAN training through theoretical analysis. However, based on the fact that research works in the field of sparse GANs are still very limited, we believe that empirical works are still needed to facilitate further theoretical study. We thank the reviewer again and we will leave the theoretical analysis to future work.
>
> > This paper is not well organized. For example, the authors present SDST for more than 1 page, which can potentially make the readers confused.
>
> `Answer 3` We thank the reviewer for pointing out the organization problem. We have updated our manuscript with the changes highlighted in a different color. More specifically: (1) we have modified section 4 to reflect the motivation of DDST better; (2) We have updated section 5 to make our proposed DDST easier to understand; (3) We have moved the training details to the appendix; (4) We have also added more detailed explanations to connect different sections better to make our work easier to read.
>
> > Some notations are abused. For example, the $\alpha$ in balance ratio and $f_\text{decay}$ should be different.
>
> `Answer 4` We thank the reviewer for pointing out the notation problem. We have replaced all $\alpha$ related to the update schedule with $\gamma$. Please see Appendix B for the change. We also go through the manuscript again and we believe there is no abused notation now.

---

### Decision · Program_Chairs · 2023-01-20

**Decision:**

Reject

**Justification For Why Not Higher Score:**

All the reviewers unanimously agree that the paper is below the accept line.

**Justification For Why Not Lower Score:**

N.A.

**Metareview: Summary, Strengths And Weaknesses:**

The manuscript proposes a novel metric to balance the dynamic sparse training of both generators and discriminators in generative neural networks (GAN), where the authors think the balance ratio plays a key role in the overfitting of GANs. The main concerns of reviewers are about: (i) novelty, while STU-GAN dynamically controls the sparsity of the generator, leaving the discriminator fixed, the authors extend this  to double dynamic sparsity by balancing the sparsity of both generator and discriminator where StyleGAN2-ADA also exploits in dense training; (ii) evaluations, where current evaluations are insufficient, at least comparisons with other typical unconditional GANs, such as BIGAN (CIFAR10 or ImageNet with low resolution) and StyleGAN2 (CIFAR10 or FFHQ); (iii) writing needs improved with clarifications on the concerns of the reviewers. In a summary, this version of manuscript is not ready for publication yet.